# Deep-Learning-Based Seismic-Signal P-Wave First-Arrival Picking Detection Using Spectrogram Images

Sugi Choi [1][ID], Bohee Lee [2][ID], Junkyeong Kim [1],*[ID] and Haiyoung Jung [1],*

1   Department of Fire and Disaster Prevention, Semyung University, 65 Semyung-ro, Jecheon-si 27136, Republic of Korea; zz55cc@naver.com
2   Department of Electrical Engineering, Semyung University, 65 Semyung-ro, Jecheon-si 27136, Republic of Korea; bhlee@semyung.ac.kr
*   Correspondence: kimjk3926@gmail.com (J.K.); hyjung@semyung.ac.kr (H.J.)

**Abstract:** The accurate detection of P-wave FAP (First-Arrival Picking) in seismic signals is crucial across various industrial domains, including coal and oil exploration, tunnel construction, hydraulic fracturing, and earthquake early warning systems. At present, P-wave FAP detection relies on manual identification by experts and automated methods using Short-Term Average to Long-Term Average algorithms. However, these approaches encounter significant performance challenges, especially in the presence of real-time background noise. To overcome this limitation, this study proposes a novel P-wave FAP detection method that employs the U-Net model and incorporates spectrogram transformation techniques for seismic signals. Seismic signals, similar to those encountered in South Korea, were generated using the stochastic model simulation program. Synthesized WGN (White Gaussian Noise) was added to replicate background noise. The resulting signals were transformed into 2D spectrogram images and used as input data for the U-Net model, ensuring precise P-wave FAP detection. In the experimental result, it demonstrated strong performance metrics, achieving an MSE of 0.0031 and an MAE of 0.0177, and an RMSE of 0.0195. Additionally, it exhibited precise FAP detection capabilities in image prediction. The developed U-Net-based model exhibited exceptional performance in accurately detecting P-wave FAP in seismic signals with varying amplitudes. Through the developed model, we aim to contribute to the advancement of microseismic monitoring technology used in various industrial fields.

**Keywords:** deep learning; FAP (First-Arrival Picking) detection; seismic signal; spectrogram images; U-Net model; WGN (white Gaussian noise)

## 1. Introduction

The essence of microseismic monitoring technology is the precise detection of P-wave FAP (first-arrival picking) for a quicker response to earthquakes [1]. Microseisms occur naturally due to tectonic or volcanic activity or are induced by activities in various industrial sectors, such as coal and oil exploration, tunnel construction, hydraulic fracturing, and geothermal power generation [2,3].

Shale gas, known for its abundant reserves and lower carbon emissions compared to traditional fossil fuels like coal and oil, has gained worldwide attention [4,5]. Hydraulic fracturing is the primary method for efficient shale gas extraction [6–8]. However, hydraulic fracturing increases ground instability and the likelihood of induced seismicity due to changes in pore pressure and variations in porous elastic stress induced by high-pressure fluid injections [9–11]. In 2017, a magnitude 5.5 earthquake occurred in Pohang, South Korea, reportedly triggered by injecting high-pressure water for geothermal power generation, leading to local ground instability [12]. Hence, real-time monitoring technology capable of swiftly and accurately detecting microseisms in industrial fields like coal and oil exploration, tunnel construction, hydraulic fracturing, and geothermal power generation

is vital. Of late, research on P-wave FAP detection in microseismic signals has gained momentum [13–16].

Both manual and automated methods are employed to detect P-wave FAP in seismic signals. Manual detection relies on the expertise of geologists, which is time-consuming, data-intensive, and subject to individual subjectivity, reducing the reliability of P-wave FAP detection [17–19]. In industrial settings, automated detection predominantly uses the STA/LTA (Short-Term Average to Long-Term Average) algorithm [20]. This method detects signal changes by calculating the ratio of the average amplitude of the input signal over a short period (STA) to that over a long period (LTA). However, STA/LTA is susceptible to background noise, blurring the boundary between signal and noise, and reducing detection performance for microseismic events with weak signals [21,22]. To address these issues and accurately detect the P-wave FAP in seismic signals even in areas with substantial background noise, various studies employing artificial intelligence for FAP automatic detection have been conducted recently [23–25]. Zhu et al. [23]. developed a model for classifying FAP points in seismic signals using various algorithms, including statistical-analysis-based DA (Discriminant Analysis), machine-learning-based logistic regression, kNN (k-Nearest Neighbor), SVM (Support Vector Machine), and a deep-learning-based CNN (Convolutional Neural Network). Although the results outperformed those of the conventional STA/LTA method, the classification accuracy of the CNN models remained limited, at 91.71%.

Recent studies have reported on the CNN-based U-Net model that is gaining significant attention for its detection performance by leveraging image features. Zhang et al. [24] proposed MT-Net using the U-Net model for multi-channel joint seismic-phase and FAP identification. The model improved the efficiency and accuracy of FAP and phase identification in seismic signals with substantial background noise by performing 2D convolution operations in U-Net based on the characteristics of sequentially arranged multi-channel data sources for P waves. Guo proposed a microseismic signal P-wave FAP detection model based on a more deeply designed network structure, U-Net++ [25]. Research based on the U-Net model added Gaussian noise to seismic simulation data to evaluate the performance of the U-Net++ model, confirming its superior performance compared to the existing STA/LTA method. However, upon analyzing the FAP detection model using the existing U-Net architecture, the findings presented in Table 1 emerged. Despite demonstrating commendable accuracy, residual errors persisted, posing a challenge in confirming detection performance through evaluation indicators like SSIM and PSNR, commonly employed in generative AI.

**Table 1.** Conventional U-Net model FAP detection performance.

| Models | Model Evaluation Index | Result | Reference |
|---|---|---|---|
| U-Net++ | MAE Accuracy | MAE = 1.21 Accuracy = 0.987 | Guo et al. [25] |
| U-Net transfer learning | Accuracy | Accuracy = 0.88 | Choi et al. [26] |
| U-Net | MSE Accuracy | MSE = 0.06 Accuracy = 0.988 | Li et al. [27] |

Therefore, this study aimed to highlight the features of microseismic signal images by applying spectrogram transformation techniques to seismic signals and incorporating WGN (white Gaussian noise). The study also aimed to develop a model for detecting the P-wave FAP in seismic signals by applying an optimized U-Net algorithm to spectrogram images. We additionally assessed the performance concerning the image luminance and signal-to-noise ratio, employing pivotal evaluation metrics like SSIM and PSNR, crucial within the domain of generative AI.

The contributions of this study are as follows:

- We synthesized seismic and WGN signals to create signal images that resemble actual images with high background noise and proposed a high-performance P-wave FAP signal-processing method using STFT (short-time Fourier transform)-based spectrogram transformation techniques.
- The P-wave FAP detection model developed in this study outperformed existing CNN and U-Net series models in terms of error, yielding an MSE of 0.0031, an MAE of 0.0177, and an RMSE of 0.0195.
- Through the developed P-wave FAP detection model, this study aimed to contribute to the advancement of microseismic monitoring technology used in various industrial fields, such as coal and oil exploration, tunnel construction, hydraulic fracturing, and earthquake early warning systems.

## 2. Development of P-Wave FAP Detection Model for Seismic Signals

Figure 1 presents the workflow of this study, and the workflow contents are as follows:

- To obtain a seismic-signal dataset, we used the SMSIM (Stochastic Model Simulation) program, considering the geological characteristics of South Korea, and generated seismic signals of various amplitudes.
- We incorporated appropriate WGN signals into the generated seismic signals and conducted a preprocessing experiment to convert the signals into spectrogram images.
- We devised a P-wave FAP detection model for seismic signals by formulating a U-Net model known for its efficacy in prior P-wave FAP detection studies, and subsequently fine-tuning the hyperparameters to enhance the model's P-wave FAP detection performance.
- To verify the reliability of the P-wave FAP detection model developed in this study, we used various model performance metrics.

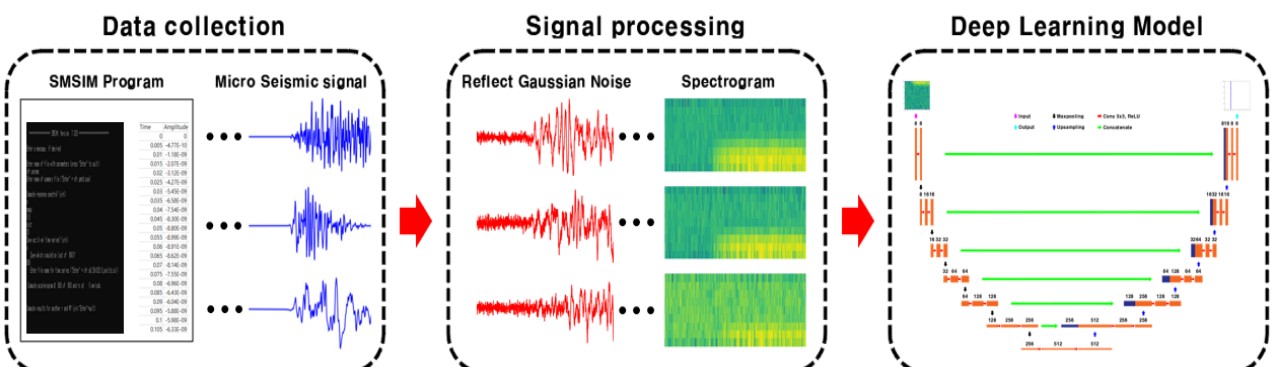

**Figure 1.** Research development process.

## 3. Experimental Details

### 3.1. Seismic-Signal Data

To collect seismic-signal data, it is necessary to install multiple sensors in the ground, and a real ground-truth dataset is essential, involving manual checking for the P-wave FAP in the seismic signals. However, training a deep-learning model requires a large dataset, and it is extremely challenging to secure enough data to accurately detect the P-wave FAP [28]. To overcome this limitation, we used the SMSIM program to generate seismic signals with intensities similar to those of earthquakes in South Korea. By incorporating the WGN signals and transforming the generated seismic signals into spectrograms, we obtained a synthetic seismic-signal dataset that mimicked real earthquakes. The synthetic seismic signals covered a range of anticipated waveform events and included signals combined with WGN noise. Table 2 displays the stress drop, quality factor, magnitude, and epicentral distance used to generate various earthquake events using the SMSIM program. Additionally, Figure 2 illustrates examples of the generated seismic signals, confirming

that the amplitude of the signal increases with the magnitude and decreases with the epicentral distance.

**Table 2.** SMSIM information.

| Parameter | Value |
|---|---|
| Stress drop | 20–200 (step: 20) |
| Q (quality factor) | 100–300 (step: 50) |
| Magnitude | 3–4 (step: 0.2) |
| Epicentral distance | 20–400 (step: 20) |
| Signal data | 10,000 (magnitude step: 5000) |

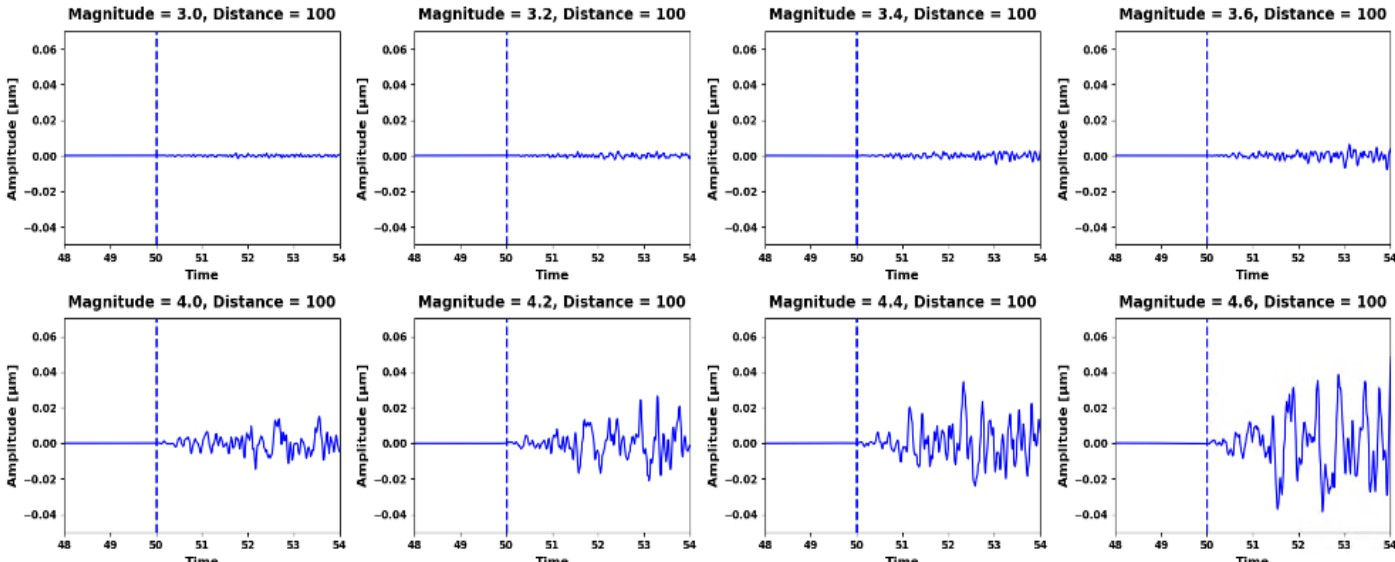

**Figure 2.** Seismic signal data.

*3.2. WGN*

WGN is a crucial concept in probability theory and signal processing; it represents noise signals following a random probability distribution. WGN is employed to model background noise occurring in many natural phenomena and has applications across various fields, such as signal and data analysis [29–31], communication system design [32], and experiments and simulations [33]. Specifically, in earthquake data processing and analysis, WGN is applied either to add noise to seismic signals or to model noise to obtain reliable results [34]. WGN follows a Gaussian distribution, with a mean of zero and a variance of $\sigma^2$. The PDF (probability density function) for this is represented in (1) [35].

$$\text{PDF}(x) = \frac{1}{\sigma\sqrt{2\pi}} \times \exp -\frac{1}{2}\left(\frac{x-\mu}{\sigma}\right)^2 \tag{1}$$

where $\text{PDF}(x)$ represents the probability density function, $\mu$ indicates the mean of the distribution, and $\sigma$ is the standard deviation of the distribution, illustrating the variability of WGN. WGN can be represented as a time-varying signal. At time t, the value of WGN can be represented as a random number extracted from a Gaussian distribution with a mean of zero and a variance of $\sigma^2$, and this value is represented in (2) [36].

$$x(t) = N\left(0, \sigma^2\right) \tag{2}$$

where $x(t)$ is the value of WGN at time t (time-series data point), and $N\left(0, \sigma^2\right)$ is the random number extracted from a Gaussian distribution with a mean of zero and a variance

of $\sigma^2$. The method of obtaining a noisy signal by adding WGN to the original signal is represented in (3) [37].

$$p(t) = s(t) + x(t) \tag{3}$$

where $p(t)$ is the value of the noisy signal at time t, and $s(t)$ is the value at time t in the original signal. This equation demonstrates how to obtain a noisy signal by combining the original signal with WGN, enabling data modeling or analysis in noisy environments. The SNR (signal-to-noise ratio) represents the power ratio of the original signal to WGN on a logarithmic scale. A high SNR value indicates that the noise is strong compared to the original signal. The SNR is one of the key metrics for signal quality. This metric can be used to analyze or evaluate the contrast between the original seismic data and the data with added noise. The theory related to the SNR is represented in (4) [38].

$$\text{SNR} = \frac{P_s}{P_n} = 10 \log_{10} \frac{\sigma_s}{\sigma_n} \tag{4}$$

where SNR is in decibels [dB], $P_s$ is the amplitude of the seismic signal, $P_n$ is the amplitude of the noise signal, $\sigma_s$ is the standard deviation of the seismic signal, and $\sigma_n$ is the standard deviation of the noise signal.

$$P = L \times \frac{1}{N} \sum_{i=0}^{N-1} |S_i|^2 \tag{5}$$

where $P$ is the total power of the signal, and $L$ is the scaling factor used to add noise to the original signal. $N$ is the number of samples in the signal, and $S_i$ represents each individual signal sample. $L$ can be used to adjust the noise level, and through $P$ and $L$, the desired SNR value can be obtained [39]. Therefore, based on the principle of the SNR, we constructed a synthetic seismic-signal dataset that reflects Gaussian noise in the actual seismic signals, making them similar to real seismic signals.

Figure 3 illustrates the WGN synthetic signals that change based on the magnitude of the original signal. As the epicenter distance decreases, the amplitude of the signal increases sharply, meaning that the original signal is influenced less by WGN. Conversely, as the epicenter distance increases, the maximum amplitude decreases and shows an average amplitude, confirming that the original signal is influenced more by WGN. Therefore, using the WGN synthetic signal data, we obtained various types of seismic-signal images, including data with clear P-wave FAPs, ambiguous data, and data with very high background noise. Figure 4 shows the signals reflecting the original and WGN when the seismic signal is large, with a magnitude of 4 and an epicenter at a distance of 20 km. As the SNRdB value increases, the amplitude of the noise also increases, generating seismic signals with relatively high background noise. Additionally, we applied SNR values of $-1$, $-5$, and $-10$ [dB] to the 10,000 preprocessed synthetic signals, generating 30,000 seismic signals.

### 3.3. Spectrogram Transformation

To analyze the features of the processed 30,000 seismic signals, we used an STFT (short-time Fourier transform)-based spectrogram transformation technique. This method visually represents the time–frequency characteristics of the signal by dividing the signal into short intervals in the time domain and applying Fourier transformations to the signals in each time interval to generate spectrograms [40,41]. Spectrograms are expressed along the time and frequency axes, enabling a clearer analysis of seismic-signal characteristics and P-wave and S-wave FAPs. They were input into CNN models to represent image features more clearly. Equation (6) describes the principle of STFT transformation [42].

$$\text{STFT}(t, \omega) = \int_{-\infty}^{\infty} x(\tau)\omega(\tau - t)exp^{-j\omega\tau} \mathrm{d}\tau \tag{6}$$

where $x(\tau)$ is the signal function, and $\omega(\tau)$ is the window function. Earthquake signals are discontinuous functions; therefore, the transformation in (7) is performed for the *nth*

discontinuous signal $x(n)$, time $m$, frequency $l$, and window function $\omega(t)$ with a window-size length $L$ [43].

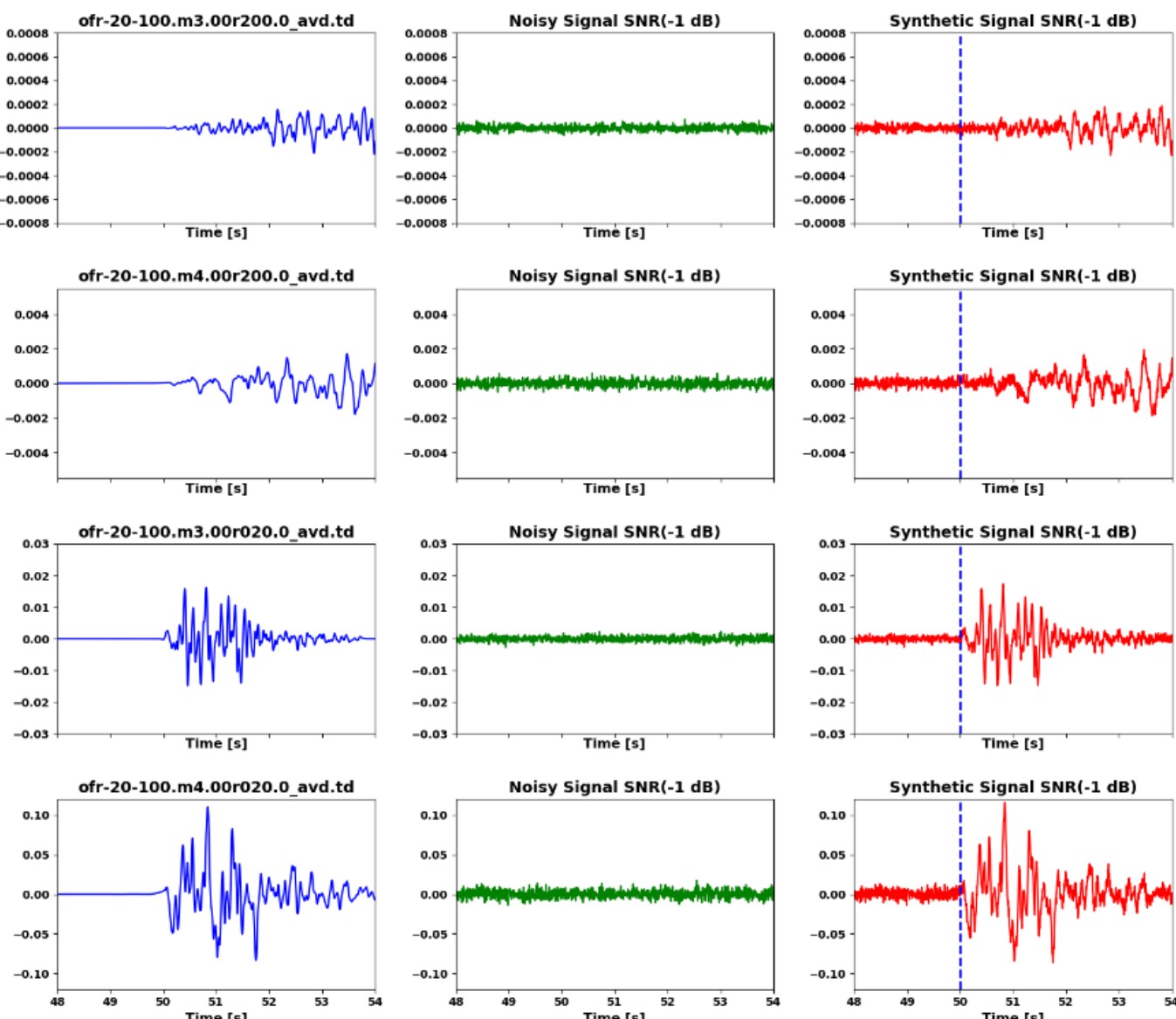

**Figure 3.** WGN synthetic signal changing according to the original signal.

$$\mathrm{STFT}(m,\, l) = \sum_{n=-\infty}^{\infty} x(n)\omega(n-m)exp^{-2\pi jnl/L} \tag{7}$$

Therefore, STFT transformation varies in frequency components depending on the size of the window function. The value of the window size has a significant impact on frequency analysis. Reducing the window size captures high-frequency components but decreases the frequency resolution, limiting the ability to analyze detailed frequency components. On the other hand, increasing the window size enhances the frequency resolution, allowing for precise frequency analysis but making it difficult to detect rapid changes in the signal. In this study, we performed experiments on spectrogram transformations based on STFT using Python's SciPy package. We applied values ranging from low to high window sizes to the seismic signals and set the overlap value to 50% to perform STFT-based spectrogram transformation. Table 3 shows the detailed spectrogram settings, such as the window size and overlap size, and the variables for WGN.

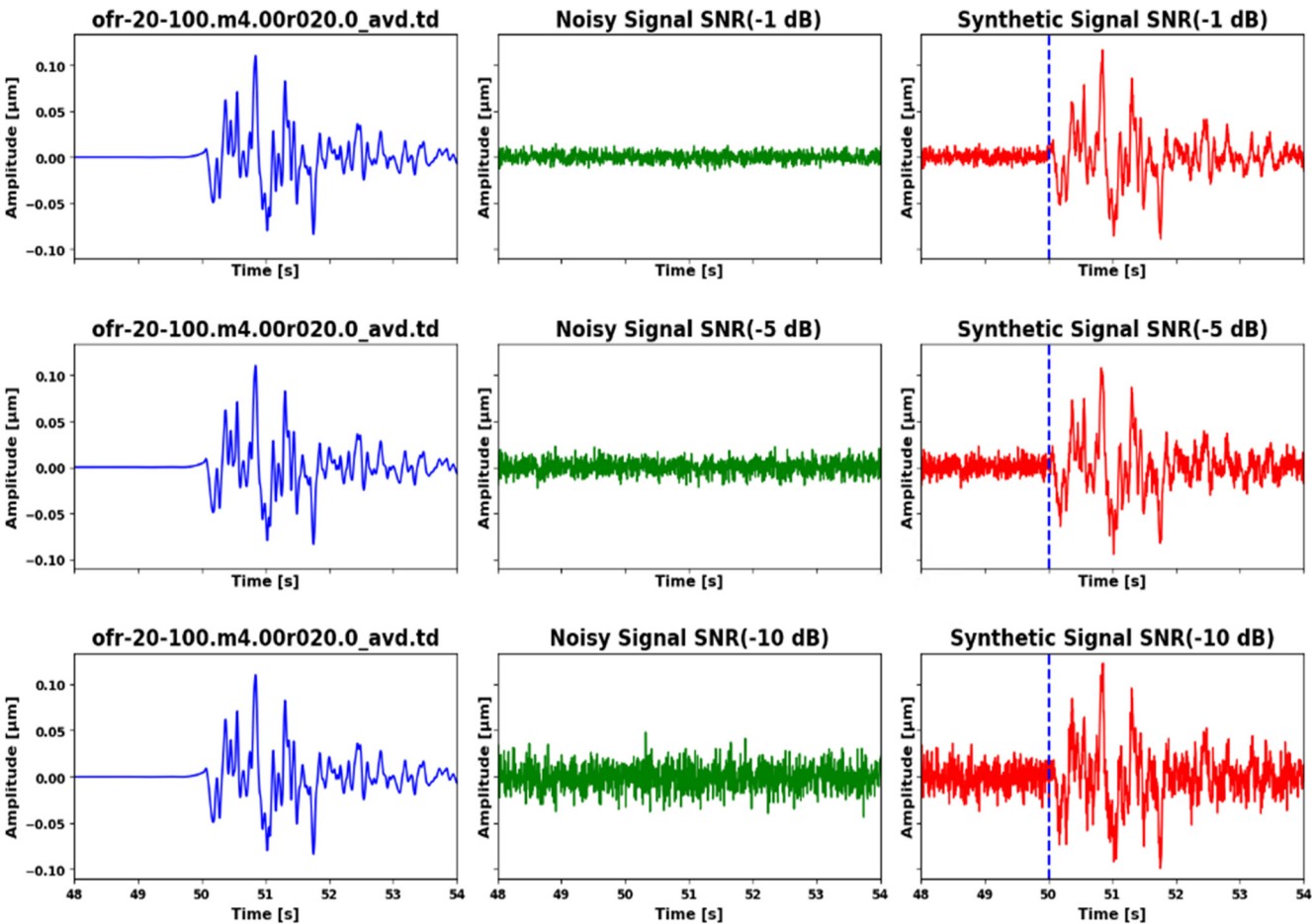

**Figure 4.** Seismic signal images reflecting WGN.

**Table 3.** Spectrogram and WGN value.

| Item | Parameter | Value |
|---|---|---|
| Spectrogram | Window size | 2, 4, 8, 16, 32, 64, 128, 256 |
| | Overlap | 1, 2, 4, 8, 16, 32, 64, 128 |
| | Recording rate | 200 |
| | Filter | Hanning window |
| White Gaussian noise | SNRdB | −1, −5, −10 |
| | Scaling factor | 1, 0.1, 0.01 |

Figure 5 illustrates the changes in spectrogram features based on window-size settings. The blue dashed line on the spectrogram image indicates that the P-wave FAP is 50 s and shows how the features of the spectrogram change based on this 50-s mark. As a result of performing the spectrogram transformation, we found that when the window size was too small, the frequency resolution decreased, making the reference points in the P-wave FAP area ambiguous. In sections with overly high window sizes, the frequency resolution increased significantly, making the distinctive points in the P-wave FAP area increasingly unclear. Therefore, we aimed to determine the optimal window size for detecting the P-wave FAP of seismic signals. We generated 30,000 spectrogram images, each with the settings specified in Table 2, and used them for model training and validation.

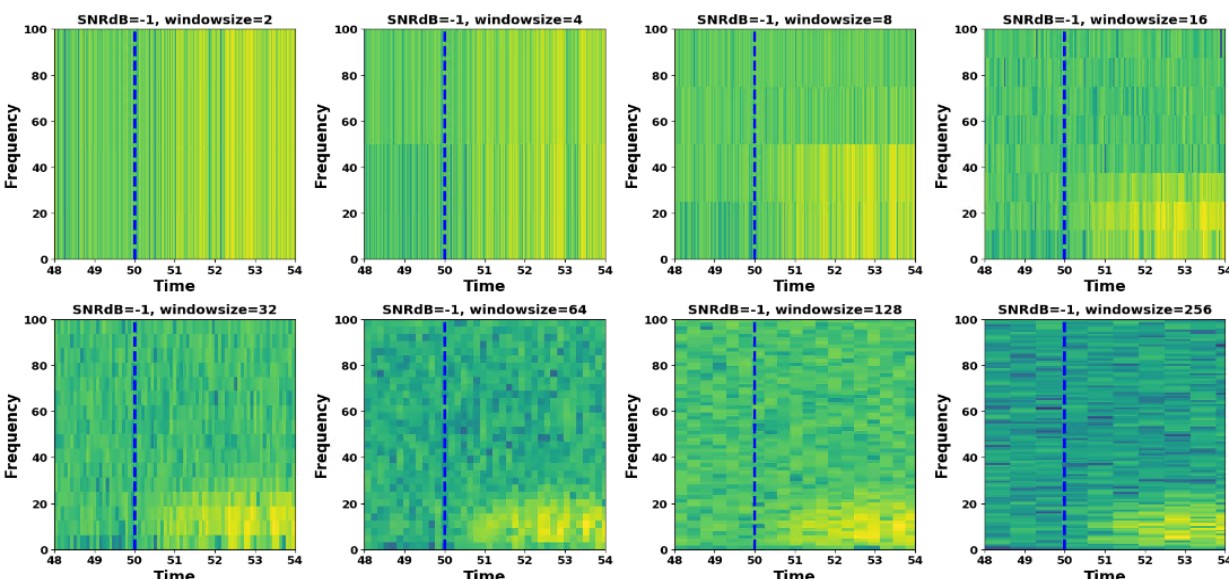

**Figure 5.** Spectrogram according to window size.

Figure 6 shows the results of the final image preprocessing. It represents the original seismic signal generated through SMSIM simulation, the spectrogram of the original signal, WGN signals, the spectrogram of WGN signals, synthetic seismic signals, the spectrogram of synthetic seismic signals, and the actual P-wave arrival points. In this study, 30,000 synthetic seismic signal spectrograms preprocessed for each window size were used as input images for the U-Net model, and 30,000 actual P wave arrival points were used as output images to generate 60,000 data points for each window size. The final images were divided into a training set of 36,000 (60%), a validation set of 12,000 (20%), and a test set of 12,000 (20%). We conducted model training and validation experiments for each window size.

*3.4. U-Net Model*

3.4.1. U-Net Framework

U-Net was first proposed in 2015 by Ronnenberger et al. [44]. It has demonstrated exceptional performance in image segmentation and has been widely used in the field of computer vision. The main architecture of U-Net consists of an encoder and a decoder. In the encoder, the input image undergoes two rounds of convolutional operations, followed by ReLU activation and downsampling. Repeating this process multiple times results in an increase in the number of channels and a reduction in the size of the feature map. In the decoder, the input from the latent space goes through upsampling, two rounds of convolutional operations, and ReLU activation. This decoding is performed multiple times, reducing the number of channels and expanding the size of the feature map as the path progresses. Additionally, the connection of feature maps between the encoder and decoder allows for fine-grained information transfer during upsampling. Such U-Net algorithms have shown excellent results in the field of seismology, including the reconstruction of seismic-signal data resolution [45,46] and P-wave FAP detection studies [47–49].

Figure 7 illustrates the U-Net architecture proposed in this study. The model consists mainly of an encoder and a decoder, both of which are designed with identical layers. The encoder is made up of convolution layers and max-pooling layers. In the convolution layer, features of the input image are extracted using a 3 × 3 filter, and the max-pooling layer performs downsampling while preserving the finer details, thereby extracting higher-level features more effectively. For precise feature extraction, we designed the convolution and max-pooling layers to be six layers deep. The decoder is composed of convolution layers and upsampling. In the convolution layer, the input image is expanded back to its original size, and its features are extracted through convolution operations. The expansion

to the original size is carried out via upsampling, which increases the feature map of the image. The process involves convolution transpose layers and concatenate layers to perform upsampling and generate the restored image at the output.

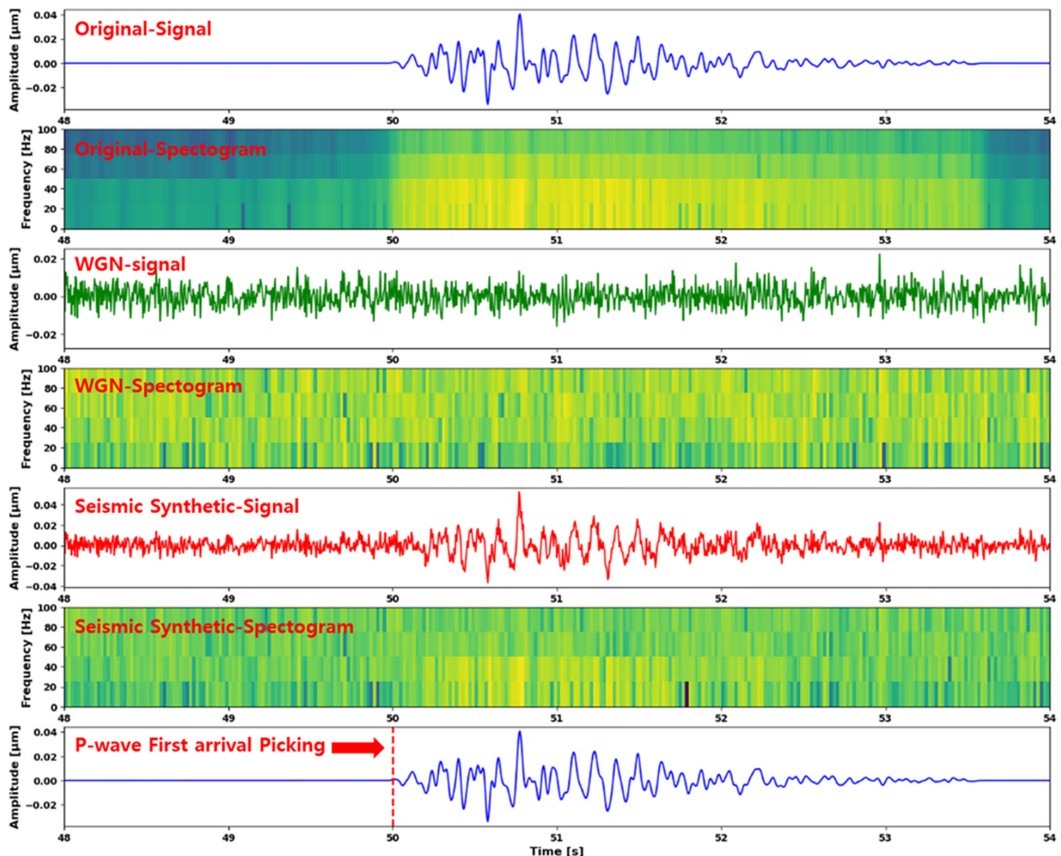

**Figure 6.** Seismic signal images.

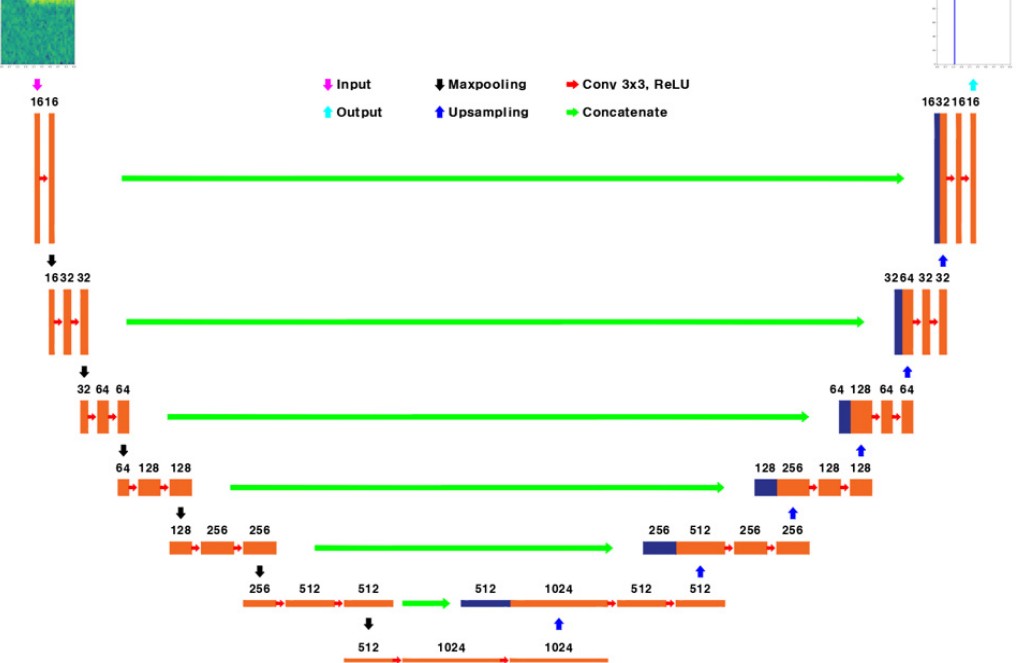

**Figure 7.** Proposed U-Net architecture.

### 3.4.2. Hyperparameter Optimization

The specific settings for optimizing the U-Net model used in this study are detailed in Table 4. The optimizer was set to Adam, and model training was conducted while adjusting the learning rate. To optimize the model, the ReduceLROnPlateau module was used to adjust the learning rate after every two epochs, and ModelCheckpoint saved the model with the highest PSNR performance. Additionally, the mean-squared error (MSE) was used as the loss function.

**Table 4.** Train parameters.

| Name of Component | Parameter | | Content and Value |
|---|---|---|---|
| Model setting value | Optimizer | | Adam |
| | Learning rate | | 0.01, 0.001, 0.0001, 0.00001 |
| | Mini-batch size | | 64 |
| | Epoch | | 100 |
| | Loss | | MSE |
| Callback | ReduceLROnPlateau | Patience | 2 |
| | | Min learning rate | 0.001 times the learning rate |
| | | Factor | 0.5 |
| | ModelCheckPoint | Best PSNR | 1 epoch |

### 3.4.3. Model Evaluation Metrics

During training, the model performance was evaluated using metrics such as the MSE, peak SRN (PSNR), and structural similarity index (SSIM).

$$\text{MSE} = \frac{1}{N} \sum_{i=1}^{n} (\hat{y}_i - y_i)^2 \tag{8}$$

where MSE is the mean-squared error [50], $N$ is the total number of data points, and $\hat{y}_i$ and $y_i$ show the actual and model-predicted values, respectively. MSE is widely used as a measure for evaluating image quality; it allows the assessment of the error between the actual and predicted images in the U-Net model [51].

$$\text{PSNR} = 10 \log_{10} \left( \frac{R^2}{\text{MSE}} \right) \tag{9}$$

where PSNR is the peak signal-to-noise ratio, and MSE is the mean-squared error. $R$ is the range of pixel values. PSNR allows for the evaluation of the loss in image quality between the actual and predicted images [51].

$$\text{SSIM}(x, y) = \frac{\left( 2\mu_x \mu_y + c_1 \right) \left( 2\sigma_{xy} + c_2 \right)}{\left( \mu^2{}_x + \mu^2{}_y + c_1 \right) \left( \sigma^2{}_x + \sigma^2{}_y + c_2 \right)} \tag{10}$$

where $x$, $y$ are the two images being compared, $\mu_x$, $\mu_y$ are the average pixel values of $x$ and $y$, $\sigma_x$, $\sigma_y$ are the variances in pixel values for $x$ and $y$, $\sigma_{xy}$ is the covariance of $x$ and $y$, and $c_1$, $c_2$ are the constants for safety. SSIM is the structural similarity index and allows for the assessment of luminance, contrast, and structure differences between the actual and predicted images [52]. Through such hyperparameter optimization and evaluation metrics, we aimed to conduct experiments for the training and performance evaluation of the U-Net model for P-wave FAP detection in seismic signals.

## 4. Experimental Results

Earlier in this paper, we outlined the generation of seismic signals, the synthesis of WGN signals, and spectrogram transformations, among other data-preprocessing tasks.

Using these, we built a dataset of 240,000 seismic signals, comprising 30,000 for each window size, and carried out training, validation, evaluation, and prediction experiments with the designed U-Net-based P-wave FAP detection model.

### 4.1. Results of U-Net Model Training and Validation

Figure 8 shows the results of the training and validation of the model by window size. As the number of epochs increases, there is a consistent trend of decreasing MSE values and increasing SSIM and PSNR scores. Notably, the best results are achieved when the window size is 64, with validations of 0.006, 0.943, and 22.296 for the MSE, SSIM, and PSNR, respectively. Therefore, we chose the spectrogram with a window size of 64 and an overlap size of 32 as the final input data.

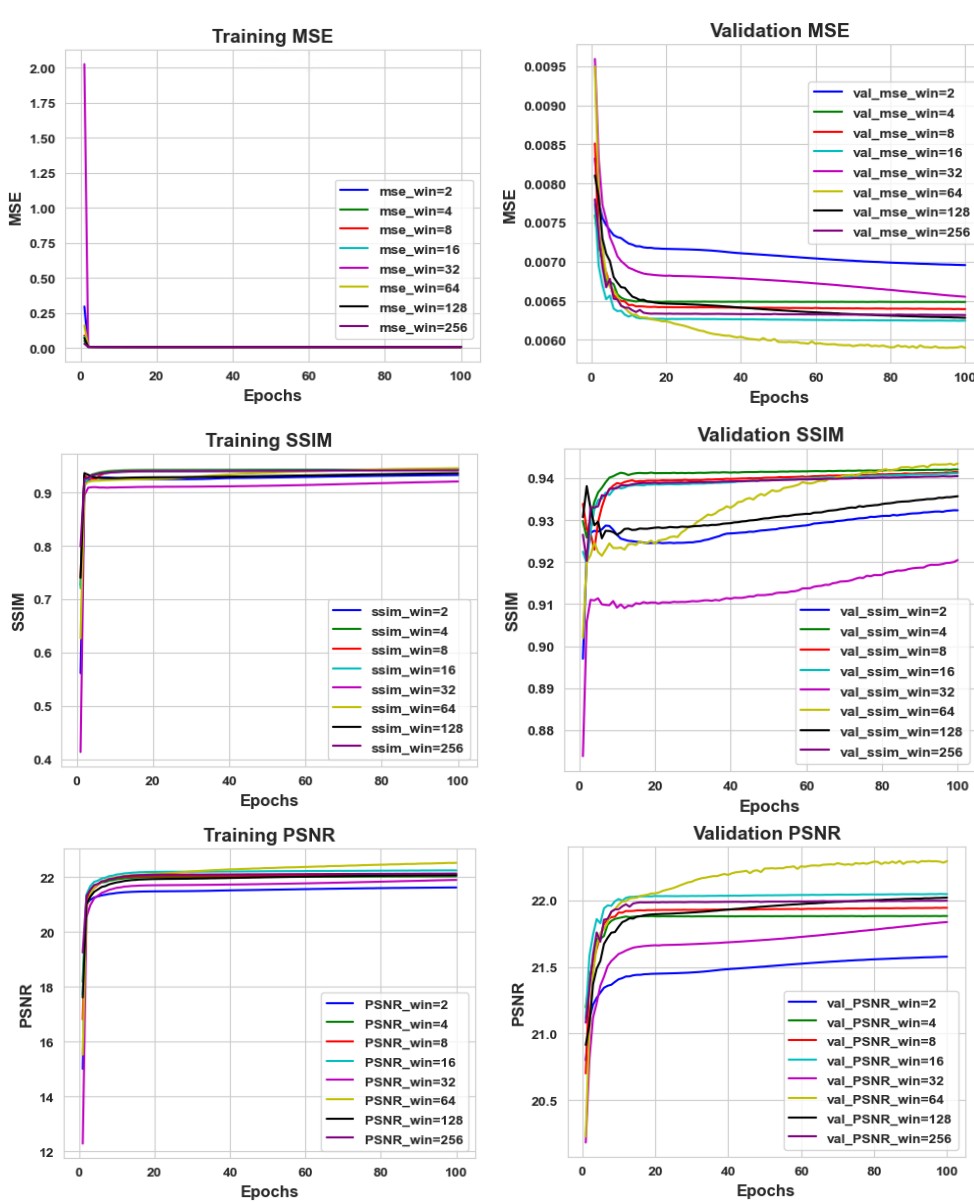

**Figure 8.** U-Net training and validation results by window size (SNRdB = −1, learning rate = 0.001).

Figure 9 shows the validation results of the model using images with a window size of 64 as the input. The developed model displays how the validation MSE, SSIM, and PSNR change per epoch for each set learning rate. Overall, the model that uses data closest to an SNRdB of −1 shows high performance. However, we confirmed that adjusting the learning rate also improved the performance of models that use data with SNRdB

of −5 and 10. Ultimately, the highest performance was observed when the SNRdB was −1 at a learning rate of 0.001, with an MSE, SSIM, and PSNR of 0.006, 0.943, and 22.23, respectively. A similarly high performance was noted for SNRdB −5 at a learning rate of 0.001, with an MSE, SSIM, and PSNR of 0.0061, 0.943, and 22.18, respectively; and for SNRdB −10 at a learning rate of 0.001, with an MSE, SSIM, and SNRdB of 0.0064, 0.942, and 21.92, respectively.

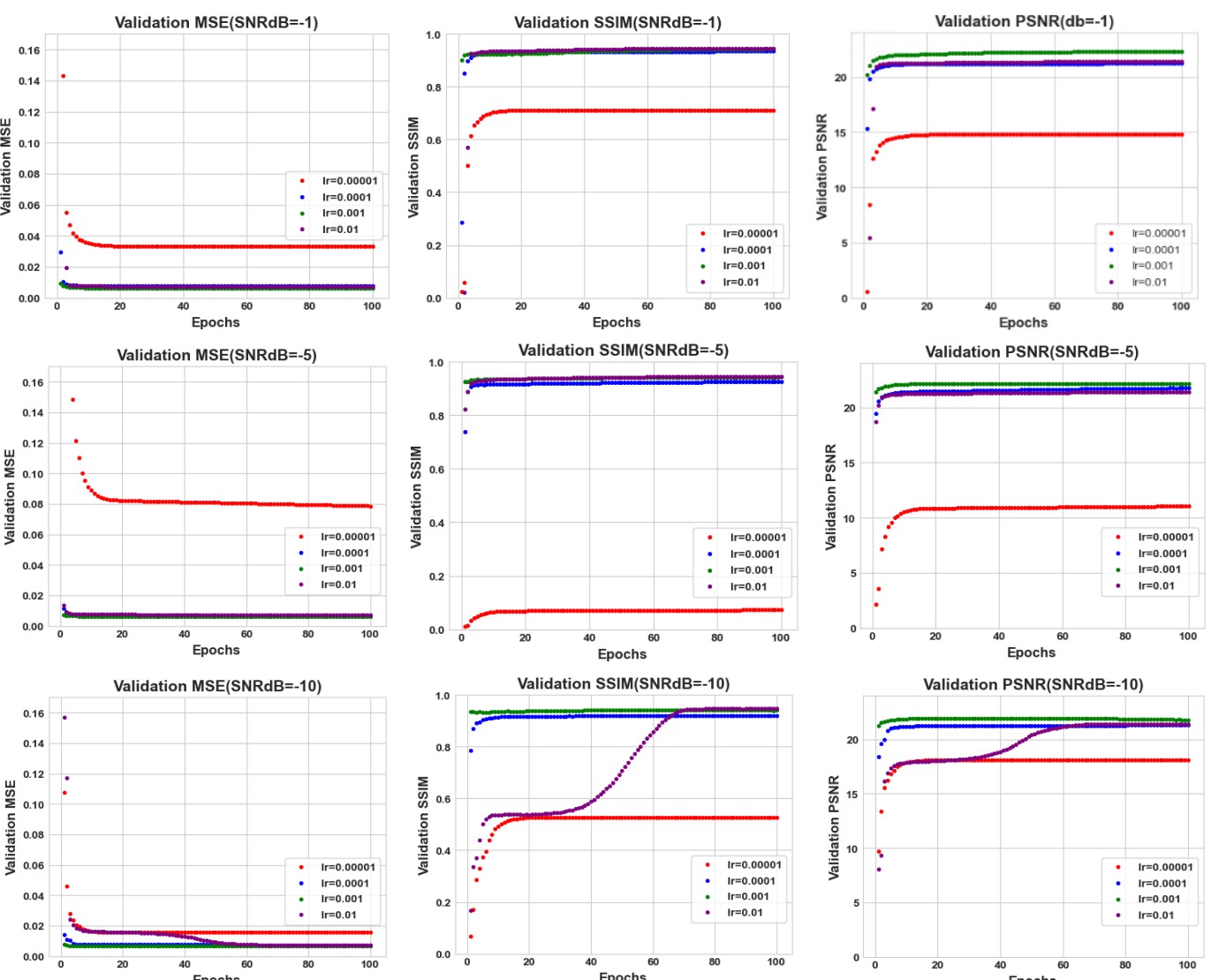

**Figure 9.** U-Net model validation results (window size = 64).

As the noise increased, the model performance declined. However, we improved the model performance through U-Net model parameter optimization based on learning rate adjustment.

### 4.2. Model Performance Evaluation and Seismic-Signal P-Wave FAP Prediction

Figure 10 displays the evaluation results of the model that showed the best performance during the validation process. The horizontal axis displays the corresponding MSE, SSIM, and PSNR values, while the vertical axis represents the count of test images associated with these values. The figure shows the evaluation results for 6000 test datasets for the models with the best performance for window size 64 and SNRdB −1, −5, and −10, as indicated in the above section, and displays the distribution of MSE, SSIM, and PSNR scores for each SNRdB. The evaluation results also show the best performance for SNRdB

of −1, consistent with the validation results. When SNRdB is −1, the values are mainly distributed at a lower MSE and higher SSIM and PSNR, while for SNRdB −5 and −10, they are relatively distributed at a higher MSE and lower SSIM and PSNR. The results exhibit a strong performance in terms of MSE and SSIM during testing. However, the testing phase yields a suboptimal performance in terms of the PSNR, which consequently leads to an overall degradation in image quality. To validate the actual predictive outcomes of these metrics, the predictive results of a representative P-wave FAP model are depicted in each of Figures 11–13.

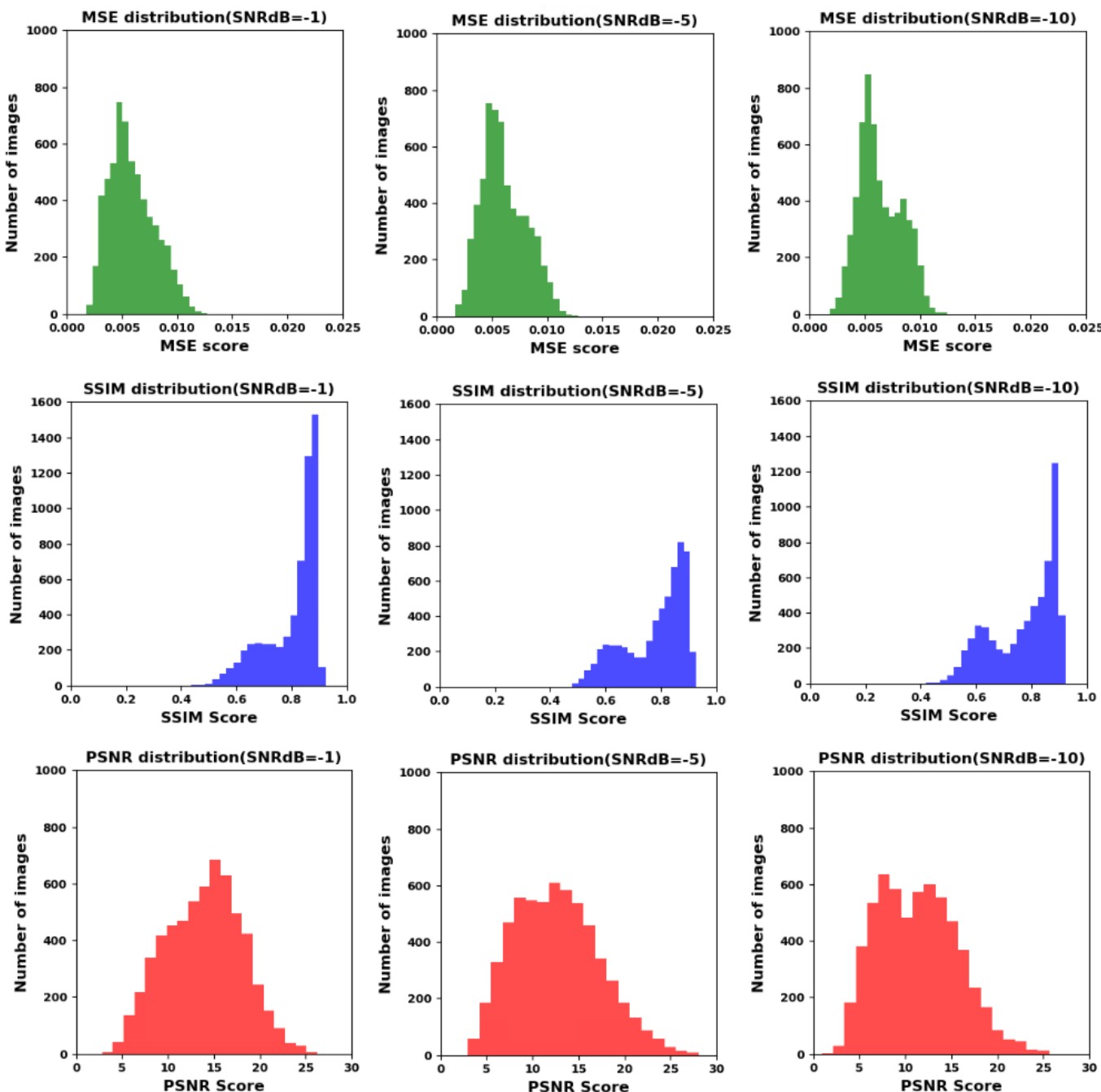

**Figure 10.** U-Net model evaluation results.

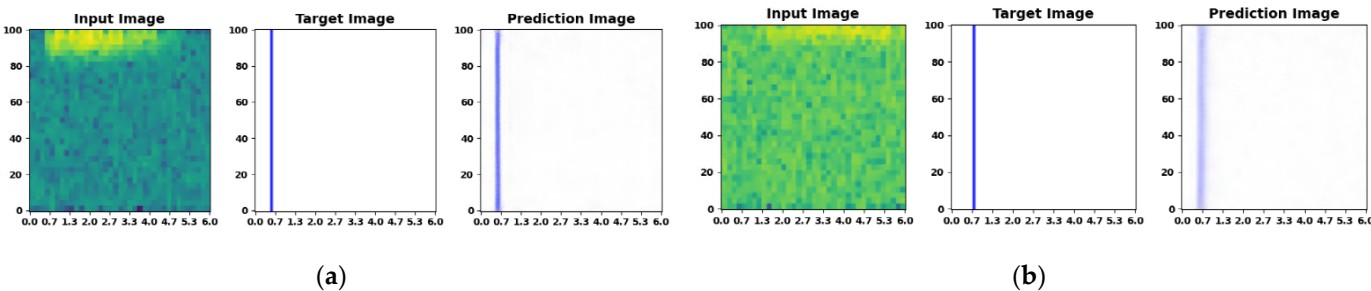

**Figure 11.** P-wave FAP prediction results when SNRdB = −1. (**a**) Input image with clear FAP. (**b**) Input image with ambiguous FAP.

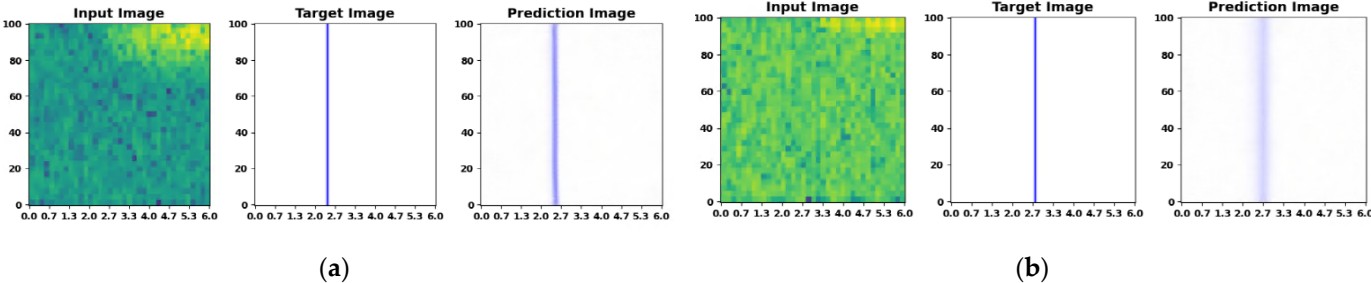

**Figure 12.** P-wave FAP prediction results when SNRdB = −5. (**a**) Input image with clear FAP. (**b**) Input image with ambiguous FAP.

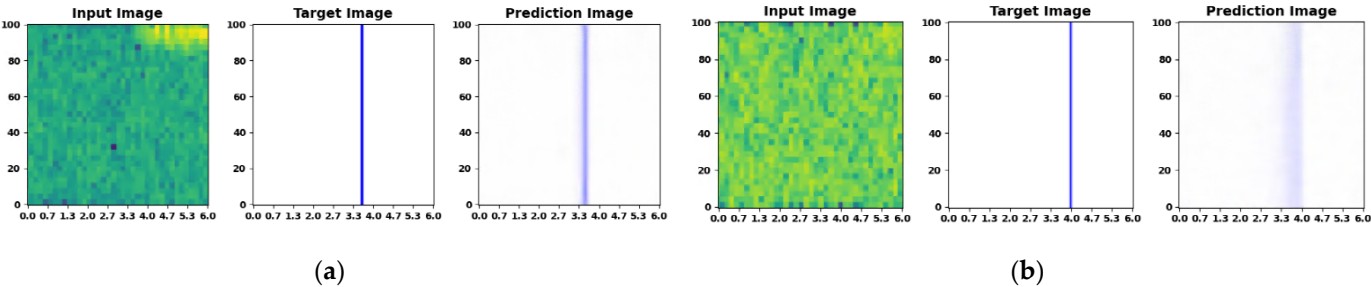

**Figure 13.** P-wave FAP prediction results when SNRdB = −10. (**a**) Input image with clear FAP. (**b**) Input image with ambiguous FAP.

The final evaluation results of the models based on SNR dB are depicted in Table 5. The numerical distributions referenced in Figure 10 have been converted into mean values, showcasing a high performance across all SNRdB values regarding error metrics such as the MSE, MAE, and RMSE. Furthermore, the accuracy, SSIM, and PSNR also exhibit high values, signifying the developed model's accurate detection capability for P-wave FAP. However, a sharp decline in SSIM and PSNR is evident specifically at an SNRdB of −10. To further analyze these findings, the predicted results derived from the images are illustrated in Figures 11–13 for each corresponding SNRdB.

**Table 5.** Model evaluation results.

| Model | SNRdB | MSE | SSIM | PSNR | MAE | RMSE | Accuracy |
|---|---|---|---|---|---|---|---|
| Window size = 64 (U-Net model) | −1 | 0.0075 | 0.9182 | 24.7937 | 0.0221 | 0.0256 | 0.9843 |
|  | −5 | 0.0031 | 0.9157 | 24.6215 | 0.0177 | 0.0195 | 0.9918 |
|  | −10 | 0.0074 | 0.7895 | 19.8422 | 0.0247 | 0.0279 | 0.9844 |

Figure 11 shows the prediction results for P-wave FAP when the SNRdB is −1. (a) shows that the predicted image is generated without error when the FAP is clear in the

input image; (b) confirms that the predicted image is generated without error, even when the FAP is relatively unclear compared to (a).

Figure 12 shows the prediction results for P-wave FAP when the SNRdB is −5. Compared to when the SNRdB is −1, the strong influence of the WGN signal on the original signal results in a deterioration of image quality in the input image, and the FAP becomes ambiguous, leading to a decline in the quality of the predicted image.

Figure 13 shows the prediction results for P-wave FAP when the SNRdB is −10. Similar to the results when the SNRdB is −5, the strong influence of the WGN signal on the original signal results in a deterioration of the image quality in the input image, and the FAP becomes ambiguous. However, the predicted images when the SNRdB was −5 and −10 were similar to the actual FAP, and there were no problems in detecting the FAP. Therefore, according to the evaluation and prediction results for the model developed in this study, although phenomena such as the degradation of image and picture quality occur, the model successfully detects the P-wave FAP of seismic signals without errors. This suggests that it could be highly effective in detecting the P-wave FAP of seismic signals with strong background noise in real-world situations.

## 5. Discussion and Comparison with Similar Works

The detection performance of seismic signals is frequently challenged by background noise. Previous studies endeavored to enhance this performance by developing a P-wave fault-arrival point (FAP) detection model through parameter and structure optimization within deep-learning frameworks. These studies predominantly utilized time-series seismic signal images or images reflecting white Gaussian noise (WGN) as inputs for U-Net model experiments. In contrast, our study introduces an advancement in terms of performance by employing 2D spectrograms as inputs. This innovative approach showcases enhanced capabilities for P-wave FAP detection. Our findings advocate for the utilization of STFT + U-Net methodology, demonstrating improved capabilities for detecting P-wave FAP. Table 6 shows a comparison of the performance of the existing and developed models.

**Table 6.** Comparison between the proposed and existing model.

| Models | Signal Processing | Model Evaluation Index | Result | Reference |
|---|---|---|---|---|
| U-Net++ | Time-series analysis of signals and WGN | MAE Accuracy | MAE = 1.21 Accuracy = 0.987 | Guo et al. [25] |
| U-Net transfer learning | Time-series analysis of signals | Accuracy | Accuracy = 0.88 | Choi et al. [26] |
| U-Net | Time-series analysis of signals and WGN | MSE Accuracy | MSE = 0.06 Accuracy = 0.988 | Li et al. [27] |
| **U-Net** | **WGN and STFT analysis** | **MSE** **MAE** **RMSE** **Accuracy** **SSIM** **PSNR** | **MSE = 0.0031** **MAE = 0.0177** **RMSE = 0.0195** **Accuracy = 0.9918** **SSIM = 0.9182** **PSNR = 24.7937** | **Our result** |

In comparative analysis with previous studies, our methodology exhibits superior performance in terms of the mean absolute error (MAE) and mean-squared error (MSE). Prior studies achieved an MAE of 1.21 [25] and an MSE of 0.06 [27], while our study achieved significantly lower values, with an MSE of 0.0031 and an MAE of 0.0177.

## 6. Conclusions

In this study, various seismic signals were generated using the SMSIM program, and a P-wave FAP detection model based on U-Net was developed, incorporating spectrogram transformation techniques and WGN signals. To obtain a dataset for model development, the SMSIM program was used to generate varying seismic signals, and WGN signals were synthesized to reflect the actual seismic signals. Subsequently, the synthesized sig-

nals were converted into image form using STFT to create spectrogram images. These converted spectrogram images served as input data for the U-Net model, and through parameter optimization and other experimental processes, a high-performance P-wave FAP detection model was developed. The developed model, when evaluated on the test dataset, successfully detected the FAP without errors between the actual and predicted images in both clean and noisy input images, although there was a degradation in image quality. Therefore, a high performance can be expected from the model developed in this study in detecting the P-wave FAP in seismic signals with strong background noise. The P-wave FAP detection model developed in this study is anticipated to be applied in various industrial sectors that require microseismic monitoring technology to predict the time and location of seismic events more accurately and contribute to improving early earthquake warning systems. Nevertheless, the decline in image quality as indicated by PSNR implies an ongoing issue pertaining to the efficacy of P-wave FAP detection. Therefore, in future research, various deep-learning models will be analyzed to enhance the P-wave FAP detection model. Additionally, in-depth research will be conducted on solutions for background noise removal, and real seismic data will be collected and analyzed to validate and improve the model's performance.

**Author Contributions:** Conceptualization, S.C. and J.K.; methodology, S.C.; software, S.C.; validation, J.K. and H.J.; formal analysis, S.C.; investigation, J.K.; resources, H.J.; data curation, B.L.; writing—original draft preparation, S.C.; writing—review and editing, B.L. and H.J.; visualization, B.L.; supervision, J.K.; project administration, H.J.; funding acquisition, H.J. All authors have read and agreed to the published version of the manuscript.

**Funding:** This work was supported by the National Research Foundation of Korea (NRF) grant funded by the Korea government (MSIT) (NRF: 2021R1G1A1014385) This research was also supported by "Regional Innovation Strategy (RIS)" through the National Research Foundation of Korea (NRF) funded by the Ministry of Education (MOE) (2021RIS–001).

**Data Availability Statement:** Data are contained within the article.

**Conflicts of Interest:** The authors declare no conflicts of interest.

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
