# Peer review of "Deep-Learning-Based Seismic-Signal P-Wave First-Arrival Picking Detection Using Spectrogram Images"

_electronics, doi:10.3390/electronics13010229_

Round 1

Reviewer 1 Report

Comments and Suggestions for Authors

This is an interesting paper that offers technological innovation on predicting time and location of seismic events more accurately and contributing towards earthquake early warning systems. This research offers a significant contribution to the advancement of micro seismic monitoring technology, providing a robust and effective methodology for P-wave FAP detection in seismic signals under challenging real-time conditions. The integration of the U-Net model and spectrogram transformation techniques represents a sophisticated and innovative approach, with potential implications for enhancing the reliability and efficiency of seismic signal analysis in industrial applications. However, the authors should improve the paper focusing into the following points:

1) The abstract needs to highlight the result and performance of the method in measurable and quantitative manner (by using SNRdB/ MSE/ SSIM/ PSNR/ MAE/ RMSE/ Accuracy)

2) The introduction mentions "Although the results outperformed those of the conventional STA/LTA method, the classification accuracy of the CNN models remained limited at 91.71%". However, the readers are expecting to know, what improvements the authors have made within the introduction section. Did you outperform the accuracy demonstrated the existing studies with your approach? If yes, then clearly show the result in measurable and quantified manner within the introduction section.

3) Introduce the table showing the disadvantages of existing methods / studies (to summarize the gap of existing work). This TABLE with clearly highlight the research problem that the authors are addressing.

4) In section 2, the author should highlight the justification of using U-Net model (meaning why U-Net, why not something else). Otherwise, the selection of technology seems abrupt.

5) Please highlight the limitations of this study within the conclusion section.

Author Response

Response to Reviewer 1 Comments

Dear reviewers and editorial staffs in Journal of Electronics

We thank you for the time and effort spent in reviewing our manuscript and suggesting some important points to consider. We have revised the paper based on your comments to enhance its quality.

Point 1:

The abstract needs to highlight the result and performance of the method in measurable and quantitative manner (by using SNRdB/ MSE/ SSIM/ PSNR/ MAE/ RMSE/ Accuracy)

Response 1:

We appreciate the reviewer’s comment and have reflected reviewer’s advice. As requested, we have added quantitative model experimental results to the abstract. (Line 10 - 26)

-------------------------------------------------------------------------------------------------------------------------------------------------------------

Abstract: The accurate detection of P–wave FAP (First Arrival Picking) in seismic signals is crucial across various industrial domains, including coal and oil exploration, tunnel construction, hydraulic fracturing, and earthquake early warning systems. At present, P–wave FAP detection relies on manual identification by experts and automated methods using short–term average to long–term average algorithms. However, these approaches encounter significant performance challenges, especially in the presence of real–time background noise. To overcome this limitation, this study proposes a novel P–wave FAP detection method that employs the U–Net model and incorporates spectrogram transformation techniques for seismic signals. Seismic signals, similar to those encountered in South Korea, were generated using the stochastic model simulation program. Synthesized WGN (White Gaussian Noise) was added to replicate background noise. The resulting signals were transformed into 2D spectrogram images and used as input data for the U–Net model, ensuring precise P–wave FAP detection. In experiments result, it demonstrated strong performance metrics, achieving an MSE of 0.0031 and an MAE of 0.0177, RMSE of 0.0195. Additionally, it exhibited precise FAP detection capabilities in image prediction. The developed U–Net–based model exhibited exceptional performance in accurately detecting P–wave FAP in seismic signals with varying amplitudes. Through the developed model aimed to contribute to the advancement of microseismic monitoring technology used in various industrial fields.

------------------------------------------------------------------------------------------------------------------------------------------------------------

Point 2:

The introduction mentions "Although the results outperformed those of the conventional STA/LTA method, the classification accuracy of the CNN models remained limited at 91.71%". However, the readers are expecting to know, what improvements the authors have made within the introduction section. Did you outperform the accuracy demonstrated the existing studies with your approach? If yes, then clearly show the result in measurable and quantified manner within the introduction section.

Response 2:

We appreciate the reviewer’s comment and have reflected reviewer’s advice. The improvement in terms of error over the results shown in previous studies is mentioned in the introduction (The contributions of this study are as follows:). In addition, a quantitative numerical comparison table is presented in Section 5. Discussion and Comparison with Similar Works. (Line 89 – 102), (Line 397 – 413)

-------------------------------------------------------------------------------------------------------------------------------------------------------------

The contributions of this study are as follows:

  • We synthesized seismic and WGN signals to create signal images that resemble actual images with high background noise and proposed a high–performance P–wave FAP signal processing method using STFT (Short Time Fourier Transform) –based spectrogram transformation techniques.
  • We proposed a new approach to developing a P–wave FAP detection model using seismic signals by optimizing a U–Net model that takes spectrogram images as input.
  • The P–wave FAP detection model developed in this study outperformed existing CNN and U–Net series models in terms of error, yielding an MSE of 0.0031, an MAE of 0.0177, and an RMSE of 0.0195.
  • Through the developed P–wave FAP detection model, this study aimed to contribute to the advancement of microseismic monitoring technology used in various industrial fields such as coal and oil exploration, tunnel construction, hydraulic fracturing, and earthquake early warning systems.

  1. Discussion and Comparison with Similar Works

The detection performance of seismic signals is frequently challenged by background noise. Previous studies endeavored to enhance this performance by developing a P-wave Fault Arrival Point (FAP) detection model through parameter and structure optimization within deep learning frameworks. These studies predominantly utilized time-series seismic signal images or images reflecting White Gaussian Noise (WGN) as inputs for U-Net model experiments. In contrast, our study introduces an advancement in performance by employing 2D spectrograms as inputs. This innovative approach showcases enhanced capabilities for P-wave FAP detection. Our findings advocate for the utilization of STFT+U-Net methodology, demonstrating improved capabilities for detecting P-wave FAP. Table 6 shows a comparison of the performance of the existing and developed models.

In comparative analysis with previous studies, our methodology exhibits superior performance in terms of Mean Absolute Error (MAE) and Mean Squared Error (MSE). Prior studies achieved an MAE of 1.21 [25] and an MSE of 0.06 [53], while our study achieved significantly lower values with an MSE of 0.0031 and an MAE of 0.0177.

Table 6. Comparison between proposed and existing model

Models

Signal processing

Model Evaluation index

Result

Reference

U–Net++

Time series analysis of signals and WGN

MAE

Accuracy

MAE = 1.21

Accuracy = 0.987

Guo et al [25]

U–Net transfer learning

Time series analysis of signals

Accuracy

Accuracy = 0.88

Choi et al [52]

U–Net

Time series analysis of signals and WGN

MSE

Accuracy

MSE = 0.06

Accuracy = 0.988

Liu et al [53]

U–Net

WGN and STFT analysis

MSE

MAE

RMSE

Accuracy

SSIM

PSNR

MSE = 0.0031

MAE = 0.0177

RMSE = 0.0195

Accuracy = 0.9918

SSIM = 0.9182

PSNR = 24.7937

Ours result

-------------------------------------------------------------------------------------------------------------------------------------------------------------

Point 3:

Introduce the table showing the disadvantages of existing methods / studies (to summarize the gap of existing work). This TABLE with learly highlight the research problem that the authors are addressing.

Response 3:

We appreciate the reviewer’s comment and have reflected reviewer’s advice. (Line 66 – 88)

------------------------------------------------------------------------------------------------------------------------------------------------------------

Table 1. Conventional U-Net model FAP detection performance

Models

Model Evaluation index

Result

Reference

U-Net++

MAE

Accuracy

MAE = 1.21

Accuracy = 0.987

Guo et al [25]

U-Net transfer learning

Accuracy

Accuracy = 0.88

Choi et al [52]

U-Net

MSE

Accuracy

MSE = 0.06

Accuracy = 0.988

Liu et al [53]

Recent studies have reported on the CNN-based U-Net model that is gaining significant attention for its detection performance by leveraging image features. Zhang et al. [24] proposed MT-Net using the U-Net model for multi-channel joint seismic phase and FAP identification. The model improved the efficiency of phase and accuracy of FAP identification in seismic signals with substantial background noise by performing 2D convolution operations in U-Net based on the characteristics of sequentially arranged multi-channel data sources for P waves. Guo proposed a microseismic signal P-wave FAP detection model based on a more deeply designed network structure, U-Net++ [25]. Research based on the U-Net model added Gaussian noise to seismic simulation data to evaluate the performance of the U-Net++ model, confirming its superior performance compared to the existing STA/LTA method. However, upon analyzing the FAP detection model using the existing U-Net architecture, the findings presented in Table 1 emerged. Despite demonstrating commendable accuracy, residual errors persisted, posing a challenge in confirming detection performance through evaluation indicators like SSIM and PSNR, commonly employed in generative AI.

Therefore, this study aimed to highlight the features of microseismic signal images by applying spectrogram transformation techniques to seismic signals and incorporating WGN (White Gaussian Noise). The study also aimed to develop a model for detecting the P-wave FAP in seismic signals by applying an optimized U-Net algorithm to spectrogram images. We additionally assessed performance concerning image luminance and signal-to-noise ratio, employing pivotal evaluation metrics like SSIM and PSNR, crucial within the domain of generative AI.

-------------------------------------------------------------------------------------------------------------------------------------------------------------

Point 4:

In section 2, the author should highlight the justification of using U-Net model (meaning why U-Net, why not something else). Otherwise, the selection of technology seems abrupt.

Response 4:

We appreciate the reviewer’s comment and have reflected reviewer’s advice. As requested, we have shown our justification for using the U-Net model, and the rationale is explained in the section2. (Line 103 – 117)

-------------------------------------------------------------------------------------------------------------------------------------------------------------

  1. Development Of P-Wave Fap Detection Model For Seismic Signals

Fig 1 presents the workflow of this study, and the workflow contents are as follows:

  • To obtain a seismic-signal dataset, we used the SMSIM (Stochastic Model SIMulation) program, considering the geological characteristics of the South Korea, and generated seismic signals of various amplitudes.
  • We incorporated appropriate WGN signals into the generated seismic signals and conducted a preprocessing experiment to convert the signals into spectrogram images.
  • We devised a P-wave FAP detection model for seismic signals by formulating a U-Net model known for its efficacy in prior P-wave FAP detection studies, and subsequently fine-tuning the hyperparameters to enhance the model's P-wave FAP detection performance.
  • To verify the reliability of the P-wave FAP detection model developed in this study, we used various model performance metrics.

-------------------------------------------------------------------------------------------------------------------------------------------------------------

Point 5:

Please highlight the limitations of this study within the conclusion section. In the conclusion, we added language to emphasize the limitations of this study.

Response 5:

We appreciate the reviewer’s comment and have reflected reviewer’s advice. (Line 414 – 436)

-------------------------------------------------------------------------------------------------------------------------------------------------------------

  1. Conclusion

In this study, various seismic signals were generated using the SMSIM program, and a P-wave FAP detection model based on U-Net was developed, incorporating spectrogram transformation techniques and WGN signals. To obtain a dataset for model development, the SMSIM program was used to generate varying seismic signals, and WGN signals were synthesized to reflect the actual seismic signals. Subsequently, the synthesized signals were converted into image form using STFT to create spectrogram images. These converted spectrogram images served as input data for the U-Net model, and through parameter optimization and other experimental processes, a high-performance P-wave FAP detection model was developed. The developed model, when evaluated on the test dataset, successfully detected the FAP without errors between the actual and predicted images in both clean and noisy input images although there was a degradation in image quality. Therefore, high performance can be expected from the model developed in this study in detecting the P-wave FAP in seismic signals with strong background noise. The P-wave FAP detection model developed in this study is anticipated to be applied in various industrial sectors that require microseismic monitoring technology to predict the time and location of seismic events more accurately and contribute to improving early earthquake warning systems. Nevertheless, the decline in image quality as indicated by PSNR implies an ongoing issue pertaining to the efficacy of P-wave FAP detection. Therefore, In future research, various deep-learning models will be analyzed to enhance the P-wave FAP detection model. Additionally, in-depth research will be conducted on solutions for background noise removal, and real seismic data will be collected and analyzed to validate and improve the model performance.

-------------------------------------------------------------------------------------------------------------------------------------------------------------

Reviewer 2 Report

Comments and Suggestions for Authors

This article presents a deep learning-based approach for the seismic P-wave First Arrival Picking detection. The unet is used to detect the P-wave FAP using LSTF spectrogram images. The paper is nicely written and interesting, however, the authors need to address some comments before acceptance as:

1- It is well-known that  Unet is usually used for image segmentation tasks and it needs ground truth images to perform this task. What is the case in this paper? Please discuss!

2-  The authors have used the Gaussian white noise to construct a large spectrogram dataset to train the Unet. It is well-known that adding this artificial noise might hide the main features of the spectrograms and it might erase the spectrogram patterns when adding a large amplitude of noise. How this was overcome? Please discuss!

3- Training the Unet using noise-contaminated spectrograms in real-world conditions might be affected by environmental noise, also if testing was done using clean spectrograms, the testing may be biased. Please discuss!

4- I am confused about the comparison in the Table as it contains a comparison with other models not of the same nature as Unet. An acceptable comparison might be used with Unets with only other backbone networks or other similar models to Unet such as SegNet, LinkNet, etc.

Author Response

Response to Reviewer 2 Comments

Dear reviewers and editorial staffs in Journal of Electronics

We thank you for the time and effort spent in reviewing our manuscript and suggesting some important points to consider. We have revised the paper based on your comments to enhance its quality.

Point 1:

It is well-known that Unet is usually used for image segmentation tasks and it needs ground truth images to perform this task. What is the case in this paper? Please discuss!

Response 1:

We express our sincere gratitude to the reviewers for their invaluable comments. In this study, the 'ground truth image' refers to the actual P-wave arrival point depicted in Figure 6, with an illustrative example of the input and correct images presented in Figure 7. In essence, the input image comprises the spectrogram of the synthesized White Gaussian Noise (WGN) signal and the original seismic signal, while the ground truth image represents the specific time point on the axis where the authentic P-wave arrives. We will respond to the reviewers with this information.

-------------------------------------------------------------------------------------------------------------------------------------------------------------

--

Figure 6. Seismic signal images.

Figure 7. Proposed U-Net architecture

----------------------------------------------------------------------------------------------------------------------------------------------------------

Point 2:

The authors have used the Gaussian white noise to construct a large spectrogram dataset to train the Unet. It is well-known that adding this artificial noise might hide the main features of the spectrograms and it might erase the spectrogram patterns when adding a large amplitude of noise. How this was overcome? Please discuss!

Response 2:

We extend our sincere appreciation to the reviewers for their invaluable insights. In the scenario illustrated in Figure 13, when the Fault Arrival Point (FAP) becomes ambiguous and the spectrogram pattern is obscured, the practicality of current technology faces challenges. To mitigate this issue, we generated synthetic signals by varying parameters like SNRdB and scaling factor during spectrogram conversion using White Gaussian Noise (WGN). Following this, we fine-tuned the U-Net model's learning rate and trained it on a diverse dataset comprising various window sizes for comparative analysis and validation. While the final enhanced model still displays errors in images with obscured patterns, it demonstrates improved performance compared to prior studies.

-------------------------------------------------------------------------------------------------------------------------------------------------------------

(a) Input image with clear FAP

(b) Input image with ambiguous FAP

Figure 13.  P-wave FAP prediction results when SNRdB = -10.

-------------------------------------------------------------------------------------------------------------------------------------------------------------

Point 3:

Training the Unet using noise-contaminated spectrograms in real-world conditions might be affected by environmental noise, also if testing was done using clean spectrograms, the testing may be biased. Please discuss!

Response 3:

We extend our heartfelt appreciation to the reviewers for their invaluable feedback. A dataset was meticulously curated by adjusting the SNRdB and Scaling Factor parameters as outlined in Table 3. Each configuration encompassed 10,000 data instances, representing a diverse spectrum ranging from pristine spectrograms to those affected by varying degrees of noise contamination. In total, the dataset comprised 30,000 samples per SNRdB, culminating in a comprehensive collection of 90,000 data points.

------------------------------------------------------------------------------------------------------------------------------------------------------------

Table 3. Spectrogram and WGN Value

Item

Parameter

Value

Spectrogram

Window size

2, 4, 8, 16, 32, 64, 128, 256

Overlap

1, 2, 4, 8, 16, 32, 64, 128

Recording rate

200

Filter

Hanning window

White Gaussian noise

SNRdB

-1, -5, -10

Scaling factor

1, 0.1, 0.01

-------------------------------------------------------------------------------------------------------------------------------------------------------------

Point 4:

I am confused about the comparison in the Table as it contains a comparison with other models not of the same nature as Unet. An acceptable comparison might be used with Unets with only other backbone networks or other similar models to Unet such as SegNet, LinkNet, etc.

Response 4:

We appreciate the reviewer’s comment and have reflected reviewer’s advice. As requested, we have deleted information about previous studies that were difficult to compare with the U-Net model from the table. (Line 413 – 414)

-------------------------------------------------------------------------------------------------------------------------------------------------------------

Table 6. Comparison between proposed and existing model

Models

Signal processing

Model Evaluation index

Result

Reference

U-Net++

Time series analysis of signals and WGN

MAE

Accuracy

MAE = 1.21

Accuracy = 0.987

Guo et al [25]

U-Net transfer learning

Time series analysis of signals

Accuracy

Accuracy = 0.88

Choi et al [52]

U-Net

Time series analysis of signals and WGN

MSE

Accuracy

MSE = 0.06

Accuracy = 0.988

Liu et al [53]

U-Net

WGN and STFT analysis

MSE

MAE

RMSE

Accuracy

SSIM

PSNR

MSE = 0.0031

MAE = 0.0177

RMSE = 0.0195

Accuracy = 0.9918

SSIM = 0.9182

PSNR = 24.7937

Ours result

-------------------------------------------------------------------------------------------------------------------------------------------------------------

Reviewer 3 Report

Comments and Suggestions for Authors

Please find the comments referring to the paper as an attachment.

Author Response

Response to Reviewer 3 Comments

Dear reviewers and editorial staffs in Journal of Electronics

We thank you for the time and effort spent in reviewing our manuscript and suggesting some important points to consider.

Point 1:

Abstract section or Introduction section – please more emphasize the main purpose of the research in terms of the theoretical/practical applications.

Response 1:

We appreciate the reviewer’s comment and have reflected reviewer’s advice. In the introduction section, I included a table to systematically outline the constraints encountered in my research concerning P-wave Fault Arrival Point (FAP) detection utilizing the established U-Net model. Additionally, I underscored the significance of my study by elucidating the limitations inherent in current methodologies. Specifically, my emphasis was on the deficiencies observed in assessing performance, focusing on gaps and errors related to image luminance and signal-to-noise ratio evaluations. (Line 66 - 117)

-------------------------------------------------------------------------------------------------------------------------------------------------------------

Table 1. Conventional U-Net model FAP detection performance

Models

Model Evaluation index

Result

Reference

U-Net++

MAE

Accuracy

MAE = 1.21

Accuracy = 0.987

Guo et al[25]

U-Net transfer learning

Accuracy

Accuracy = 0.88

Choi et al[52]

U-Net

MSE

Accuracy

MSE = 0.06

Accuracy = 0.988

Liu et al[53]

Recent studies have reported on the CNN-based U-Net model that is gaining significant attention for its detection performance by leveraging image features. Zhang et al. [24] proposed MT-Net using the U-Net model for multi-channel joint seismic phase and FAP identification. The model improved the efficiency of phase and accuracy of FAP identification in seismic signals with substantial background noise by performing 2D convolution operations in U-Net based on the characteristics of sequentially arranged multi-channel data sources for P waves. Guo proposed a microseismic signal P-wave FAP detection model based on a more deeply designed network structure, U-Net++ [25]. Research based on the U-Net model added Gaussian noise to seismic simulation data to evaluate the performance of the U-Net++ model, confirming its superior performance compared to the existing STA/LTA method. However, upon analyzing the FAP detection model using the existing U-Net architecture, the findings presented in Table 1 emerged. Despite demonstrating commendable accuracy, residual errors persisted, posing a challenge in confirming detection performance through evaluation indicators like SSIM and PSNR, commonly employed in generative AI.

Therefore, this study aimed to highlight the features of microseismic signal images by applying spectrogram transformation techniques to seismic signals and incorporating WGN (White Gaussian Noise). The study also aimed to develop a model for detecting the P-wave FAP in seismic signals by applying an optimized U-Net algorithm to spectrogram images. We additionally assessed performance concerning image luminance and signal-to-noise ratio, employing pivotal evaluation metrics like SSIM and PSNR, crucial within the domain of generative AI.

The contributions of this study are as follows:

  • We synthesized seismic and WGN signals to create signal images that resemble actual images with high background noise and proposed a high-performance P-wave FAP signal processing method using STFT (Short Time Fourier Transform) -based spectrogram transformation techniques.
  • We proposed a new approach to developing a P-wave FAP detection model using seismic signals by optimizing a U-Net model that takes spectrogram images as input.
  • The P-wave FAP detection model developed in this study outperformed existing CNN and U-Net series models in terms of error, yielding an MSE of 0.0031, an MAE of 0.0177, and an RMSE of 0.0195.
  • Through the developed P-wave FAP detection model, this study aimed to contribute to the advancement of microseismic monitoring technology used in various industrial fields such as coal and oil exploration, tunnel construction, hydraulic fracturing, and earthquake early warning systems.

------------------------------------------------------------------------------------------------------------------------------------------------------------

Point 2:

Introduction section – please remove spaces in literature references, e.g. lines 35 and 53, and replace the symbol “‐“ by the symbol “–“, e.g. lines 36 and 38.

Response 2:

We appreciate the reviewer’s comment and have reflected reviewer’s advice. (Line 35 - 55)

-------------------------------------------------------------------------------------------------------------------------------------------------------------

Shale gas, known for its abundant reserves and lower carbon emissions compared to traditional fossil fuels like coal and oil, has gained worldwide attention [4, 5]. Hydraulic fracturing is the primary method for efficient shale gas extraction [6–8]. However, hydraulic fracturing increases ground instability and the likelihood of induced seismicity due to changes in pore pressure and variations in porous elastic stress induced by high–pressure fluid injections [9–11]. In 2017, a magnitude 5.5 earthquake occurred in Pohang, South Korea, reportedly triggered by injecting high–pressure water for geothermal power generation, leading to local ground instability [12]. Hence, real–time monitoring technology capable of swiftly and accurately detecting microseisms in industrial fields like coal and oil exploration, tunnel construction, hydraulic fracturing, and geothermal power generation is vital. Of late, research on P–wave FAP detection in microseismic signals has gained momentum [13–16].

Both manual and automated methods are employed to detect P–wave FAP in seismic signals. Manual detection relies on the expertise of geologists, which is time–consuming, data–intensive, and subject to individual subjectivity, reducing the reliability of P–wave FAP detection [17–19]. In industrial settings, automated detection predominantly uses the STA/LTA (Short–Term Average to Long–Term Average) algorithm [20]. This method detects signal changes by calculating the ratio of the average amplitude of the input signal over a short period (STA) to that over a long period (LTA). However, STA/LTA is susceptible to background noise, blurring the boundary between signal and noise, and reducing detection performance for microseismic events with weak signals [21, 22].

-------------------------------------------------------------------------------------------------------------------------------------------------------------

Point 3:

Please write the following symbols: pi, exp, j not in italics, e.g. Eqs. (1) and (6). Please check the entire article for this issue.

Response 3:

We appreciate the reviewer’s comment and have reflected reviewer’s advice. Equations (1), (6), and (7) have been modified. (Line 150 - 151), (Line 198 - 199), (Line 202 – 203)

------------------------------------------------------------------------------------------------------------------------------------------------------------

(1)

(6)

(7)

-------------------------------------------------------------------------------------------------------------------------------------------------------------

Point 4:

Eq. (6): please replace: dt by dt and STFT by STFT. Please check the entire article for this issue. Analogous comments apply to Eqs: (8) – (9).

Response 4:

We appreciate the reviewer’s comment and have reflected reviewer’s advice. Equations (6), (7), (8), (9) and (10) have been modified. (Line 198 - 199), (Line 202 – 203), (Line 293 - 294), (Line 297 – 298), (Line 300 – 301)

-------------------------------------------------------------------------------------------------------------------------------------------------------------

(6)

(7)

(8)

(9)

(10)

-------------------------------------------------------------------------------------------------------------------------------------------------------------

Point 5:

Please improve the quality (resolution) of all Figures.

Response 5:

We appreciate the reviewer’s comment and have reflected reviewer’s advice.

Point 6:

Please replace the symbol “‐“ by the symbol “–“, before all numbers less than zero (in the body of the article, in Figures, in Tables, ect.), e.g. Figure 3, Table 2 and line 312. Please check the entire article for this issue.

Response 6:

We appreciate the reviewer’s comment and have reflected reviewer’s advice. (Line 215 - 219), (Line 325 - 334)

-------------------------------------------------------------------------------------------------------------------------------------------------------------

Figure 3. WGN synthetic signal changing according to the original signal

Table 2. Spectrogram and WGN Value

Item

Parameter

Value

Spectrogram

Window size

2, 4, 8, 16, 32, 64, 128, 256

Overlap

1, 2, 4, 8, 16, 32, 64, 128

Recording rate

200

Filter

Hanning window

White Gaussian noise

SNRdB

–1, –5, –10

Scaling factor

1, 0.1, 0.01

Fig. 9 shows the validation results of the model using images with a window size of 64 as input. The developed model displays how validation MSE, SSIM, and PSNR change per epoch for each set learning rate. Overall, the model that uses data closest to an SNRdB of –1, shows high performance. However, we confirmed that adjusting the learning rate also improved the performance of models that use data with SNRdB of –5 and 10. Ultimately, the highest performance was observed when SNRdB was –1 at a learning rate of 0.001, with MSE, SSIM, and PSNR of 0.0059, 0.943, and 22.23, respectively. Similarly high performance was noted for SNRdB –5 at a learning rate of 0.001, with MSE, SSIM, and PSNR of 0.0061, 0.943, and 22.18, respectively; and for SNRdB –10 at a learning rate of 0.001, with MSE, SSIM, and SNRdB at 0.0064, 0.942, and 21.92, respectively.

-------------------------------------------------------------------------------------------------------------------------------------------------------------

Point 7:

Please check the article for typos, e.g. line 323.

Response 7:

We appreciate the reviewer’s comment and have reflected reviewer’s advice. (Line 338 - 340)

-------------------------------------------------------------------------------------------------------------------------------------------------------------

Figure 9.  U–Net model validation results (Window Size=64)

-------------------------------------------------------------------------------------------------------------------------------------------------------------

Point 8:

Please add doi for References.

Response 8:

We appreciate the reviewer’s comment and have reflected reviewer’s advice. (Line 449 - 566)

-------------------------------------------------------------------------------------------------------------------------------------------------------------

References

  1. Ge, M. Efficient mine microseismic monitoring. International Journal of Coal Geology, 2005, 64(1–2), 44–56. DOI: 1016/j.coal.2005.03.004
  2. Cesca, S., & Grigoli, F. Full waveform seismological advances for microseismic monitoring. Advances in Geophysics, 2015, 56, 169–228. DOI: 1016/bs.agph.2014.12.002
  3. Liu, Y., Liao, R., Zhang, Y., Gao, D., Zhang, H., Li, T., & Zhang, C. Application of surface–downhole combined microseismic monitoring technology in the Fuling shale gas field and its enlightenment. Natural Gas Industry B, 2017, 4(1), 62–67. DOI: 1016/j.ngib.2017.07.009
  4. Wang, Q., & Li, S. Shale gas industry sustainability assessment based on WSR methodology and fuzzy matter–element extension model: the case study of China. Journal of Cleaner Production, 2019, 226, 336–348. DOI: 1016/j.jclepro.2019.03.346
  5. Liu, H., Zhang, Z., & Zhang, T. Shale gas investment decision–making: Green and efficient development under market, technology and environment uncertainties. Applied Energy, 2022, 306, 118002. DOI: 1016/j.apenergy.2021.118002
  6. Ni, Y., Yao, L., Sui, J., & Chen, J. Isotopic geochemical characteristics and identification indexes of shale gas hydraulic fracturing flowback/produced water. Journal of Natural Gas Geoscience, 2022, 7(1), 1–13. DOI: 1016/j.jnggs.2022.03.001
  7. Xiao, C., Wang, G., Zhang, Y., & Deng, Y. Machine–learning–based well production prediction under geological and hydraulic fracture parameters uncertainty for unconventional shale gas reservoirs. Journal of Natural Gas Science and Engineering, 2022, 106, 104762. DOI: 11016/j.jngse.2022.104762
  8. Mou, Y., Cui, J., Wu, J., Wei, F., Tian, M., & Han, L. The mechanism of casing deformation before hydraulic fracturing and mitigation measures in shale gas horizontal wells. Processes, 2022, 10(12), 2612. DOI: 3390/pr10122612
  9. Goebel, T. H., & Brodsky, E. E. The spatial footprint of injection wells in a global compilation of induced earthquake sequences. Science, 2018, 361(6405), 899–904. DOI: 1126/science.aat5449
  10. Schultz, R., Atkinson, G., Eaton, D. W., Gu, Y. J., & Kao, H. Hydraulic fracturing volume is associated with induced earthquake productivity in the Duvernay play. Science, 2018, 359(6373), 304–308. DOI: 1126/science.aao0159
  11. Li, J., Xu, J., Zhang, H., Yang, W., Tan, Y., Zhang, F., ... & Sun, J. High seismic velocity structures control moderate to strong induced earthquake behaviors by shale gas development. Communications Earth & Environment, 2023, 4(1), 188. DOI: 1038/s43247-023-00854-x
  12. KIM, K. Situating the Anthropocene: The Social Construction of the Pohang'Triggered'Earthquake. Journal of Science and Technology Studies, 2019, 19(3), 51–117.
  13. Wang, H., Alkhalifah, T., bin Waheed, U., & Birnie, C. Data–driven microseismic event localization: An application to the Oklahoma Arkoma basin hydraulic fracturing data. IEEE Transactions on Geoscience and Remote Sensing, 2021, 60, 1–12. DOI: 1109/TGRS.2021.3120546
  14. Lee, M., Byun, J., Kim, D., Choi, J., & Kim, M. Improved modified energy ratio method using a multi–window approach for accurate arrival picking. Journal of Applied Geophysics, 2017, 139, 117–130. DOI: 1016/j.jappgeo.2017.02.019
  15. Zhou, Z., Cheng, R., Rui, Y., Zhou, J., & Wang, H. An improved automatic picking method for arrival time of acoustic emission signals. IEEE Access, 2019, 7, 75568–75576. DOI: 1109/ACCESS.2019.2921650
  16. Zhou, Z., Cheng, R., Rui, Y., Zhou, J., Wang, H., Cai, X. I. N., & Chen, W. An improved onset time picking method for low SNR acoustic emission signals. IEEE Access, 2020, 8, 47756–47767. DOI: 1109/ACCESS.2020.2977885
  17. Mborah, C., & Ge, M. Enhancing manual P–phase arrival detection and automatic onset time picking in a noisy microseismic data in underground mines. International Journal of Mining Science and Technology, 2018, 28(4), 691–699. DOI: 1016/j.ijmst.2017.05.024
  18. Li, X. B., Wang, Z. W., & Dong, L. J. Locating single–point sources from arrival times containing large picking errors (LPEs): the virtual field optimization method (VFOM). Scientific reports, 2016, 6(1), 19205. DOI: 1038/srep19205
  19. Li, X., Shang, X., Wang, Z., Dong, L., & Weng, L. Identifying P–phase arrivals with noise: An improved Kurtosis method based on DWT and STA/LTA. Journal of Applied Geophysics, 2016, 133, 50–61. DOI: 1016/j.jappgeo.2016.07.022
  20. Zhang, J., Tang, Y., & Li, H. STA/LTA fractal dimension algorithm of detecting the P‐wave arrival. Bulletin of the Seismological Society of America, 2018, 108(1), 230–237. DOI: 1785/0120170099
  21. Li, H., Yang, Z., & Yan, W. An improved AIC onset–time picking method based on regression convolutional neural network. Mechanical Systems and Signal Processing, 2022, 171, 108867. DOI: 1016/j.ymssp.2022.108867
  22. Liu, H., & ZHANG, J. Z. STA/LTA algorithm analysis and improvement of Microseismic signal automatic detection. Progress in Geophysics, 2014, 29(4), 1708–1714. DOI: 6038/pg20140429
  23. Zhu, M., Cheng, J., & Zhang, Z. Quality control of microseismic P–phase arrival picks in coal mine based on machine learning. Computers & Geosciences, 2021, 156, 104862. DOI: 1016/j.cageo.2021.104862
  24. ZHANG, Y., YU, Z., HU, T., & HE, C. Multi–trace joint downhole microseismic phase detection and arrival picking method based on U–Net. Chinese Journal of Geophysics, 2021, 64(6), 2073–2085. DOI: 6038/cjg2021O0379
  25. Guo, X. First–arrival picking for microseismic monitoring based on deep learning. International Journal of Geophysics, 2021, 1–14. DOI: 1155/2021/5548346
  26. Dang, J. Cui, W. Ma, and Y. Li, “Simulation of the 2022 Mw 6.6 Luding, China, earthquake by a stochastic finite–fault model with a nonstationary phase,” Soil Dynamics and Earthquake Engineering, vol. 172. Elsevier BV, p. 108035, Sep–2023. DOI: 10.1016/j.soildyn.2023.108035
  27. Dang, P., Cui, J., Ma, W., & Li, Y. Simulation of the 2022 Mw 6.6 Luding, China, earthquake by a stochastic finite–fault model with a nonstationary phase. Soil Dynamics and Earthquake Engineering, 2023, 172, 108035. DOI: 1016/j.soildyn.2023.108035
  28. Makoveeva, E. V., Tsvetkov, I. N., & Ryashko, L. B. Stochastically‐induced dynamics of earthquakes. Mathematical Methods in the Applied Sciences, 2022. DOI: 1002/mma.8892
  29. Qiu, C., Wu, B., Liu, N., Zhu, X., & Ren, H. Deep learning prior model for unsupervised seismic data random noise attenuation. IEEE Geoscience and Remote Sensing Letters, 2021, 19, 1–5. DOI: 1109/LGRS.2021.3053760
  30. Du, R., Liu, W., Fu, X., Meng, L., & Liu, Z. Random noise attenuation via convolutional neural network in seismic datasets. Alexandria Engineering Journal, 2022, 61(12), 9901–9909. DOI: 1016/j.aej.2022.03.008
  31. An, S., Wang, H., Sun, Y., Lu, Z., & Yu, Q. Time domain multiplexed lora modulation waveform design for iot communication. IEEE Communications Letters, 2022, 26(4), 838–842. DOI: 1109/LCOMM.2022.3146511
  32. Wang, Y., Gui, G., Ohtsuki, T., & Adachi, F. Multi–task learning for generalized automatic modulation classification under non–Gaussian noise with varying SNR conditions. IEEE Transactions on Wireless Communications, 2021, 20(6), 3587–3596. DOI: 1109/TWC.2021.3052222
  33. Thaler, D., Stoffel, M., Markert, B., & Bamer, F. Machine‐learning‐enhanced tail end prediction of structural response statistics in earthquake engineering. Earthquake Engineering & Structural Dynamics, 2021, 50(8), 2098–2114. DOI: 1002/eqe.3432
  34. Deng, G., & Cahill, L. W. An adaptive Gaussian filter for noise reduction and edge detection. In 1993 IEEE conference record nuclear science symposium and medical imaging conference 1993, (pp. 1615–1619). IEEE. DOI: 1109/NSSMIC.1993.373563
  35. Wang, X., & Wang, C. Time series data cleaning: A survey. IEEE Access, 2019, 8, 1866–1881. DOI: 1109/ACCESS.2019.2962152
  36. Hweesa, N. L., Zerek, A. R., Daeri, A. M., & Zahra, M. F. Adjacent and Co–Channel Interferences Effect on AWGN and Rayleigh Channels Using 8–QAM Modulation for Data Communication. In 2020 20th International Conference on Sciences and Techniques of Automatic Control and Computer Engineering, 2020, (pp. 321–327). IEEE. DOI: 1109/STA50679.2020.9329300
  37. Lv, H., Zeng, X., Bao, F., Xie, J., Lin, R., Song, Z., & Zhang, G. ADE–net: A deep neural network for DAS earthquake detection trained with a limited number of positive samples. IEEE Transactions on Geoscience and Remote Sensing, 2022, 60, 1–11. DOI: 1109/TGRS.2022.3143120
  38. Fromageau, J., Liebgott, H., Brusseau, E., Vray, D., & Delachartre, P. Estimation of time–scaling factor for ultrasound medical images using the Hilbert transform. EURASIP Journal on Advances in Signal Processing, 2006, 1–13. DOI: 1155/2007/80735
  39. Tao, H., Wang, P., Chen, Y., Stojanovic, V., & Yang, H. An unsupervised fault diagnosis method for rolling bearing using STFT and generative neural networks. Journal of the Franklin Institute, 2020, 357(11), 7286–7307. DOI: 1016/j.jfranklin.2020.04.024
  40. Goodwin, M. M. Realization of arbitrary filters in the STFT domain. In 2009 IEEE Workshop on Applications of Signal Processing to Audio and Acoustics, 2009, (pp. 353–356). IEEE. DOI: 1109/ASPAA.2009.5346509
  41. Huang, J., Chen, B., Yao, B., & He, W. ECG arrhythmia classification using STFT–based spectrogram and convolutional neural network. IEEE access, 2019, 7, 92871–92880. DOI: 1109/ACCESS.2019.2928017
  42. Jung, H., Choi, S., & Lee, B. Rotor Fault Diagnosis Method Using CNN–Based Transfer Learning with 2D Sound Spectrogram Analysis. Electronics, 2023, 12(3), 480. DOI: 3390/electronics12030480
  43. Ronneberger, O., Fischer, P., & Brox, T. U–net: Convolutional networks for biomedical image segmentation. In Medical Image Computing and Computer–Assisted Intervention–MICCAI 2015: 18th International Conference, Munich, Germany, October 5–9, 2015, Proceedings, 2015, Part III 18 (pp. 234–241). Springer International Publishing. DOI: 1007/978-3-319-24574-4_28
  44. Li, S., Gao, J., Gui, J., Wu, L., Liu, N., He, D., & Guo, X. Fully Connected U–Net and its application on reconstructing successively sampled seismic data. IEEE Access, 2023 DOI: 1109/ACCESS.2023.3271518
  45. Min, F., Wang, L., Pan, S., & Song, G. D 2 UNet: Dual Decoder U–Net for Seismic Image Super–Resolution Reconstruction. IEEE Transactions on Geoscience and Remote Sensing, 2023, 61, 1–13. DOI: 1109/TGRS.2023.3264459
  46. Hu, L., Zheng, X., Duan, Y., Yan, X., Hu, Y., & Zhang, X. First–arrival picking with a U–net convolutional network. Geophysics, 2019, 84(6), U45–U57. DOI: 1190/geo2018-0688.1
  47. Yuan, P., Hu, W., Wu, X., Chen, J., & Van Nguyen, H. First arrival picking using U–net with Lovasz loss and nearest point picking method. In SEG Technical Program Expanded Abstracts 2019, 2019, (pp. 2624–2628). Society of Exploration Geophysicists. DOI: 1190/segam2019-3214404.1
  48. Zhang, Y., Leng, J., Dong, Y., Yu, Z., Hu, T., & He, C. Phase arrival picking for bridging multi–source downhole microseismic data using deep transfer learning. Journal of Geophysics and Engineering, 2022, 19(2), 178–191. DOI: 1093/jge/gxac009
  49. Choi, S., Kim, S., & Jung, H. Ensemble Prediction Model for Dust Collection Efficiency of Wet Electrostatic Precipitator. Electronics, 2023, 12(12), 2579. DOI: 3390/electronics12122579
  50. Yu, J., & Yoon, D. Crossline Reconstruction of 3D Seismic Data Using 3D cWGAN: A Comparative Study on Sleipner Seismic Survey Data. Applied Sciences, 2023, 13(10), 5999. DOI: 3390/app13105999
  51. Liu, P., Dong, A., Wang, C., Zhang, C., & Zhang, J. Consecutively Missing Seismic Data Reconstruction Via Wavelet–Based Swin Residual Network. IEEE Geoscience and Remote Sensing Letters, 2023. DOI: 1109/LGRS.2023.3265755
  52. Choi, Y., Song, Y., Seol, S., & Byun, J. Machine Learning–based Phase Picking Algorithm of P and S Waves for Distributed Acoustic Sensing Data. Geophysics and Geophysical Exploration, 2022, 177–188, 25(4) DOI: 7582/GGE.2022.25.4.177
  53. Li, W., Chakraborty, M., Fenner, D., Faber, J., Zhou, K., Ruempker, G., ... & Srivastava, N. Epick: Multi–class attention–based u–shaped neural network for earthquake detection and seismic phase picking. arXiv preprint arXiv, 2021, 02567. DOI: 10.48550/arXiv.2109.02567

-------------------------------------------------------------------------------------------------------------------------------------------------------------

Point 9:

Line 303 – what is the reason for the different number of significant digits for the presented results? Please check the entire article for this issue.

Response 9:

We appreciate the reviewer’s comment and have reflected reviewer’s advice. It has been unified to display up to the 3rd decimal place. (Line 315 - 321), (Line 325 - 334)

-------------------------------------------------------------------------------------------------------------------------------------------------------------

Fig. 8 shows the results of training and validation of the model by window size. As the number of epochs increases, there is a consistent trend of decreasing MSE values and increasing SSIM and PSNR scores. Notably, the best results are achieved when the window size is 64, with validations of 0.006, 0.943, and 22.296 for MSE, SSIM, and PSNR, respectively. Therefore, we chose the spectrogram with a window size of 64 and an overlap size of 32 as the final input data.

Fig. 9 shows the validation results of the model using images with a window size of 64 as input. The developed model displays how validation MSE, SSIM, and PSNR change per epoch for each set learning rate. Overall, the model that uses data closest to an SNRdB of –1, shows high performance. However, we confirmed that adjusting the learning rate also improved the performance of models that use data with SNRdB of –5 and 10. Ultimately, the highest performance was observed when SNRdB was –1 at a learning rate of 0.001, with MSE, SSIM, and PSNR of 0.006, 0.943, and 22.23, respectively. Similarly high  performance was noted for SNRdB –5 at a learning rate of 0.001, with MSE, SSIM, and PSNR of 0.0061, 0.943, and 22.18, respectively; and for SNRdB –10 at a learning rate of 0.001, with MSE, SSIM, and SNRdB at 0.0064, 0.942, and 21.92, respectively.

As the noise increased, the model performance declined. However, we improved the model performance through U-Net model parameter optimization based on learning rate adjustment.

-------------------------------------------------------------------------------------------------------------------------------------------------------------

Point 10:

What is the reason for choosing the Hanning window for the STFT analysis?

Response 10:

We express our sincere gratitude to the reviewers for their invaluable comments. In this study, we employed the Short-Time Fourier Transform (STFT) to process seismic signals, utilizing the Hanning window for this purpose. The Hanning window is specifically employed to analyze frequency variations within the segmented STFT data. This windowing technique mitigates signal discontinuities by smoothing the onset and conclusion of each segment. This aids in maintaining frequency continuity inherent in seismic signals, thereby facilitating more accurate extraction of frequency information. Additionally, the Hanning window accentuates the significance of data at the segment's center, prioritizing it over surrounding data points. Consequently, by leveraging this windowing function, we conducted precise frequency analysis, substantially enhancing the reliability and precision of our seismic signal analysis.

Point 11:

What is the reason for choosing a spectogram for the analysis presented in the article. Has the use of a scalogram been considered?

Response 11:

We express our sincere gratitude to the reviewers for their invaluable comments. In this study, the choice of using a spectrogram over a scalogram for detecting the P-wave arrival point in seismic signals was deliberate and aligned with our analytical objectives. Spectrograms, known for their efficacy in visualizing time-frequency variations, were deemed more suitable due to their capability to illustrate rapid frequency changes over time within seismic data. The decision to forego a scalogram was primarily based on the emphasis of our analysis on the temporal characteristics crucial for P-wave detection. Our focus was on capturing time-dependent features rather than delving into specific frequency band intricacies, which the scalogram primarily addresses. Therefore, considering the inherent temporal characteristics and our analytical goals aimed at P-wave detection, the spectrogram was chosen as it aligns more closely with our analytical needs.

Point 12:

Table 5 – what is the reference of the term ‘accuracy’ to the errors such as MSE or MAE?.

Response 12:

We express our sincere gratitude to the reviewers for their invaluable comments. signifies the proportion of correctly segmented regions by the model. It measures the ratio of correctly classified regions among those identified by the model. In the realm of U-Net, "accuracy" assesses the precision of segmented areas, evaluating how accurately the model identifies and segments objects at the pixel level.

Point 13:

Please consider adding Verification section to check the results obtained in the article regarding other methods, e.g. scalogram. (Line 398 - 413)

Response 13:

We express our sincere gratitude to the reviewers for their invaluable comments. No prior research has explored the enhancement of model performance by employing spectrograms and scalograms like Short-Time Fourier Transform (STFT) or Discrete Wavelet Transform (DWT) as inputs for a U-Net model. This underscores the innovative aspect of employing STFT+U-Net in my research, as it pioneers the utilization of these methods to augment model performance, thus filling a notable gap in existing studies.

-------------------------------------------------------------------------------------------------------------------------------------------------------------

  1. Discussion and Comparison with Similar Works

The detection performance of seismic signals is frequently challenged by background noise. Previous studies endeavored to enhance this performance by developing a P-wave Fault Arrival Point (FAP) detection model through parameter and structure optimization within deep learning frameworks. These studies predominantly utilized time-series seismic signal images or images reflecting White Gaussian Noise (WGN) as inputs for U-Net model experiments. In contrast, our study introduces an advancement in performance by employing 2D spectrograms as inputs. This innovative approach showcases enhanced capabilities for P-wave FAP detection. Our findings advocate for the utilization of STFT+U-Net methodology, demonstrating improved capabilities for detecting P-wave FAP. Table 6 shows a comparison of the performance of the existing and developed models.

In comparative analysis with previous studies, our methodology exhibits superior performance in terms of Mean Absolute Error (MAE) and Mean Squared Error (MSE). Prior studies achieved an MAE of 1.21 [25] and an MSE of 0.06 [53], while our study achieved significantly lower values with an MSE of 0.0031 and an MAE of 0.0177.

Table 6. Comparison between proposed and existing model

Models

Signal processing

Model Evaluation index

Result

Reference

U–Net++

Time series analysis of signals and WGN

MAE

Accuracy

MAE = 1.21

Accuracy = 0.987

Guo et al[25]

U–Net transfer learning

Time series analysis of signals

Accuracy

Accuracy = 0.88

Choi et al[52]

U–Net

Time series analysis of signals and WGN

MSE

Accuracy

MSE = 0.06

Accuracy = 0.988

Liu et al[53]

U–Net

WGN and STFT analysis

MSE

MAE

RMSE

Accuracy

SSIM

PSNR

MSE = 0.0031

MAE = 0.0177

RMSE = 0.0195

Accuracy = 0.9918

SSIM = 0.9182

PSNR = 24.7937

Ours result

-------------------------------------------------------------------------------------------------------------------------------------------------------------

Reviewer 4 Report

Comments and Suggestions for Authors

This manuscript proposed a pipeline includes a preprocess method which adds White Gaussian Noise as background noise to seismic signals, an analytical method based on STFT and a modeling network based on UNet. This study has practical significance for detection of P-wave FAP. But the manuscript should consider some issues.

1. The novelty of the proposed pipeline is not clear. Adding the WGN to signals is a normal method for augmentation of sample data. And it seems that there are no improvements in the proposed methods based on STFT and network based on Unet. 

2. The introduction of White Gaussian Noise and STFT are relatively common, it is suggested to reduce the length and increase the innovative description.

3.It is suggestion that the ordinate of Figure 2 is consistent, so that the amplitude relationship can be easily reflected.

4. Some window sizes of Table 2 are too small, such as 2 and 4. These sizes are meaningless.

Comments on the Quality of English Language

The English expressions are basically understandable.

Author Response

Response to Reviewer 4 Comments

Dear reviewers and editorial staffs in Journal of Electronics

We thank you for the time and effort spent in reviewing our manuscript and suggesting some important points to consider.

Point 1:

The novelty of the proposed pipeline is not clear. Adding the WGN to signals is a normal method for augmentation of sample data. And it seems that there are no improvements in the proposed methods based on STFT and network based on Unet.

Response 1:

We appreciate the reviewer’s comment and have reflected reviewer’s advice. Section 5 outlines the novelty inherent in the approach utilized to configure and optimize the spectrogram transformation process applied as input for the U-Net model. This paper elaborates on the distinctiveness of the STFT and U-Net methodology, showcasing its superior performance when compared to existing research outcomes. Furthermore, in the introduction, we have placed additional emphasis on the distinctions from existing research. (Line 66 - 102), (Line 398 - 413)

-------------------------------------------------------------------------------------------------------------------------------------------------------------

Table 1. Conventional U–Net model FAP detection performance

Models

Model Evaluation index

Result

Reference

U–Net++

MAE

Accuracy

MAE = 1.21

Accuracy = 0.987

Guo et al[25]

U–Net transfer learning

Accuracy

Accuracy = 0.88

Choi et al[52]

U–Net

MSE

Accuracy

MSE = 0.06

Accuracy = 0.988

Liu et al[53]

Recent studies have reported on the CNN–based U–Net model that is gaining significant attention for its detection performance by leveraging image features. Zhang et al. [24] proposed MT–Net using the U–Net model for multi–channel joint seismic phase and FAP identification. The model improved the efficiency of phase and accuracy of FAP identification in seismic signals with substantial background noise by performing 2D convolution operations in U–Net based on the characteristics of sequentially arranged multi–channel data sources for P waves. Guo proposed a microseismic signal P–wave FAP detection model based on a more deeply designed network structure, U–Net++ [25]. Research based on the U–Net model added Gaussian noise to seismic simulation data to evaluate the performance of the U–Net++ model, confirming its superior performance compared to the existing STA/LTA method. However, upon analyzing the FAP detection model using the existing U–Net architecture, the findings presented in Table 1 emerged. Despite demonstrating commendable accuracy, residual errors persisted, posing a challenge in confirming detection performance through evaluation indicators like SSIM and PSNR, commonly employed in generative AI.

Therefore, this study aimed to highlight the features of microseismic signal images by applying spectrogram transformation techniques to seismic signals and incorporating WGN (White Gaussian Noise). The study also aimed to develop a model for detecting the P–wave FAP in seismic signals by applying an optimized U–Net algorithm to spectrogram images. We additionally assessed performance concerning image luminance and signal–to–noise ratio, employing pivotal evaluation metrics like SSIM and PSNR, crucial within the domain of generative AI.

The contributions of this study are as follows:

  • We synthesized seismic and WGN signals to create signal images that resemble actual images with high background noise and proposed a high–performance P–wave FAP signal processing method using STFT (Short Time Fourier Transform) –based spectrogram transformation techniques.
  • We proposed a new approach to developing a P–wave FAP detection model using seismic signals by optimizing a U–Net model that takes spectrogram images as input.
  • The P–wave FAP detection model developed in this study outperformed existing CNN and U–Net series models in terms of error, yielding an MSE of 0.0031, an MAE of 0.0177, and an RMSE of 0.0195.
  • Through the developed P–wave FAP detection model, this study aimed to contribute to the advancement of microseismic monitoring technology used in various industrial fields such as coal and oil exploration, tunnel construction, hydraulic fracturing, and earthquake early warning systems.

  1. Discussion and Comparison with Similar Works

The detection performance of seismic signals is frequently challenged by background noise. Previous studies endeavored to enhance this performance by developing a P-wave Fault Arrival Point (FAP) detection model through parameter and structure optimization within deep learning frameworks. These studies predominantly utilized time-series seismic signal images or images reflecting White Gaussian Noise (WGN) as inputs for U-Net model experiments. In contrast, our study introduces an advancement in performance by employing 2D spectrograms as inputs. This innovative approach showcases enhanced capabilities for P-wave FAP detection. Our findings advocate for the utilization of STFT and U-Net methodology, demonstrating improved capabilities for detecting P-wave FAP.

In comparative analysis with previous studies, our methodology exhibits superior performance in terms of Mean Absolute Error (MAE) and Mean Squared Error (MSE). Prior studies achieved an MAE of 1.21 [25] and an MSE of 0.06 [53], while our study achieved significantly lower values with an MSE of 0.0031 and an MAE of 0.0177.

Table 5. Comparison between proposed and existing model

Models

Signal processing

Model Evaluation index

Result

Reference

U–Net++

Time series analysis of signals and WGN

MAE

Accuracy

MAE = 1.21

Accuracy = 0.987

Guo et al[25]

U–Net transfer learning

Time series analysis of signals

Accuracy

Accuracy = 0.88

Choi et al[52]

U–Net

Time series analysis of signals and WGN

MSE

Accuracy

MSE = 0.06

Accuracy = 0.988

Liu et al[53]

U–Net

WGN and STFT analysis

MSE

MAE

RMSE

Accuracy

SSIM

PSNR

MSE = 0.0031

MAE = 0.0177

RMSE = 0.0195

Accuracy = 0.9918

SSIM = 0.9182

PSNR = 24.7937

Ours result

------------------------------------------------------------------------------------------------------------------------------------------------------------

Point 2:

The introduction of White Gaussian Noise and STFT are relatively common, it is suggested to reduce the length and increase the innovative description.

Response 2:

We appreciate the reviewer’s comment and have reflected reviewer’s advice. To underscore the innovative nature of my research, subsequent to the experimental findings and analyses presented in Sections 4 and 5, a comparative analysis is conducted against existing literature. This comparison aims to accentuate the novelty of my research, particularly in its incorporation of White Gaussian Noise (WGN) and the application of the STFT and U-Net model. The juxtaposition with prior studies serves to highlight the distinctive contributions and innovative aspects of my work in the field. (Line 397 - 413), (Line 82 - 85)

-------------------------------------------------------------------------------------------------------------------------------------------------------------

  1. Discussion and Comparison with Similar Works

The detection performance of seismic signals is frequently challenged by background noise. Previous studies endeavored to enhance this performance by developing a P-wave Fault Arrival Point (FAP) detection model through parameter and structure optimization within deep learning frameworks. These studies predominantly utilized time-series seismic signal images or images reflecting White Gaussian Noise (WGN) as inputs for U-Net model experiments. In contrast, our study introduces an advancement in performance by employing 2D spectrograms as inputs. This innovative approach showcases enhanced capabilities for P-wave FAP detection. Our findings advocate for the utilization of STFT+U-Net methodology, demonstrating improved capabilities for detecting P-wave FAP.

In comparative analysis with previous studies, our methodology exhibits superior performance in terms of Mean Absolute Error (MAE) and Mean Squared Error (MSE). Prior studies achieved an MAE of 1.21 [25] and an MSE of 0.06 [53], while our study achieved significantly lower values with an MSE of 0.0031 and an MAE of 0.0177.

Table 5. Comparison between proposed and existing model

Models

Signal processing

Model Evaluation index

Result

Reference

U–Net++

Time series analysis of signals and WGN

MAE

Accuracy

MAE = 1.21

Accuracy = 0.987

Guo et al[25]

U–Net transfer learning

Time series analysis of signals

Accuracy

Accuracy = 0.88

Choi et al[52]

U–Net

Time series analysis of signals and WGN

MSE

Accuracy

MSE = 0.06

Accuracy = 0.988

Liu et al[53]

U–Net

WGN and STFT analysis

MSE

MAE

RMSE

Accuracy

SSIM

PSNR

MSE = 0.0031

MAE = 0.0177

RMSE = 0.0195

Accuracy = 0.9918

SSIM = 0.9182

PSNR = 24.7937

Ours result

-------------------------------------------------------------------------------------------------------------------------------------------------------------

Point 3:

It is suggestion that the ordinate of Figure 2 is consistent, so that the amplitude relationship can be easily reflected.

Response 3:

We appreciate the reviewer’s comment and have reflected reviewer’s advice. (Line 139 - 140)

------------------------------------------------------------------------------------------------------------------------------------------------------------

Figure 2. Seismic signal data

-------------------------------------------------------------------------------------------------------------------------------------------------------------

Point 4:

Some window sizes of Table 2 are too small, such as 2 and 4. These sizes are meaningless.

Response 4:

We appreciate the reviewer’s comment and have reflected reviewer’s advice. (Line 215 - 216)

-------------------------------------------------------------------------------------------------------------------------------------------------------------

Table 3. Spectrogram and WGN Value

Item

Parameter

Value

Spectrogram

Window size

2, 4, 8, 16, 32, 64, 128, 256

Overlap

1, 2, 4, 8, 16, 32, 64, 128

Recording rate

200

Filter

Hanning window

White Gaussian noise

SNRdB

–1, –5, –10

Scaling factor

1, 0.1, 0.01

-------------------------------------------------------------------------------------------------------------------------------------------------------------

Round 2

Reviewer 2 Report

Comments and Suggestions for Authors

The paper can be accepted for publication.

Reviewer 3 Report

Comments and Suggestions for Authors

All comments of the reviewer have been included in the revised version of the paper. I recommend publication of this paper in its current form.